# Classifying compound coastal storm and heavy rainfall events in the north-western Spanish Mediterranean

Marc Sanuy[1], Tomeu Rigo[2], José A. Jiménez[1], M. Carmen Llasat[3]

[1]Laboratori d'Enginyeria Marítima, Universitat Politècnica de Catalunya, BarcelonaTech, c/Jordi Girona 1-3, Campus Nord ed. D1, Barcelona, 08034, Spain
[2]Servei Meteorològic de Catalunya, C. Berlin, 38-46, 08029, Barcelona, Spain
[3]GAMA, Department of Applied Physics, University of Barcelona, Barcelona, 08028, Spain

*Correspondence to*: Marc Sanuy (marc.sanuy@upc.edu), José A Jiménez (jose.jimenez@upc.edu)

**Abstract.** The Northwest (NW) Mediterranean coastal zone is a populous and well-developed area in which the impact of natural hazards like flash floods and coastal storms can result in frequent and significant damages. Although the occurrence and impacts of such hazards have been widely covered, few studies have considered their combined impact on the region, which would result in more damage. Within this context, this study analyses the occurrence and characteristics of compound extreme events of heavy rainfall episodes (as a proxy for flash floods) and coastal storms (using the maximum significant wave height) along the Catalan coast as a paradigm of the NW Mediterranean. Two different types of events are considered: multivariate, in which the two hazards occur at the same location, and spatially compounding, in which they occur within the same limited time window and their impacts accumulate at distinct and separate locations. The analysis is regionally performed along a coastline extension of about 600 km by considering seven coastal sectors and their corresponding river catchment basins. Once the compound events are analysed, the synoptic atmospheric pressure fields are analysed to determine the prevailing weather conditions that generated them. Finally, a Bayesian network is used to fully characterise these events over the territory. The obtained results show that the NW Mediterranean, represented by the Catalan coast, has a high probability of experiencing compound extreme events. Despite the relatively small size of the study area, there are significant variations in the event characteristics along the territory, with the most frequent type being spatially compound, except in the northernmost sectors where multivariate events dominate. These northern sectors also present the highest correlation in the intensity of both hazards. Three representative synoptic situations have been identified as dominant for the occurrence of these events, with different relative importance levels of the compounding drivers (rainfall and waves) and different distributions of impacts across coastal basins. Overall, results obtained from specific events indicated that heavy rainfall is related to the most significant impacts despite having a larger spatial reach .

## 1. Introduction

Coastal zones are one of the highest risk areas in the world given the concentration of natural hazards, people, and buildings along coastlines (e.g. Kron, 2013). Among the different hazards, flooding is currently the most frequent, dangerous, and costly (IPCC, 2012; Blöschl et al., 2020), and it is very likely to significantly increase under climate change (e.g. Hallegatte, 2013; IPCC, 2014; Alfieri, 2015; Blöschl, 2017). One of the intrinsic characteristics of flooding in coastal areas is that it can be induced by different climatic drivers such as storm surge, run-up, rainfall, and/or river flow, each of which may act individually but are often interconnected (Berghuijs et al., 2019). Moreover, when flooding is induced by marine drivers, such as storm surge and/or waves, impacting sedimentary coastlines, erosion also occurs simultaneously. Thus, although risk assessments in coastal zones usually consider the impact of sea hazards and climate drivers individually (e.g. Michaelides et al., 2018; Van Dongeren et al., 2018), they should instead be considered as the result of compounding events (Hao et al., 2018; Ward et al., 2018). In this sense, an increasing number of studies have stressed the importance of compound flooding in coastal zones at

different geographical scales (Wahl et al., 2015; Wu et al., 2018; Bevacqua et al., 2019; Hendry et al., 2019), including their potential increase under the influence of climate change (Moftakhari et al., 2017; Bevacqua et al., 2019). When the importance of these types of events is considered across Europe, the Mediterranean coastline can be considered a hotspot. On the one hand, more than 50% of its population is concentrated in the coastal zone, increasing the risk to human life due to flooding (Vinet et al., 2019). On the other hand, the relative frequency of flash floods in the region is the highest in Europe (Gaume et al., 2016), and impacts related to climate and environmental changes are more severe relative to the global average, with temperatures already reaching +1.5 ℃ relative to pre-industrial times (Cramer et al., 2018). This combination also implies an increase in coastal storm–induced damage over the last decades (e.g. Jiménez et al., 2012; Garnier et al. 2018). However, there are a limited number of studies assessing the combined effect of different hazard-inducing climate drivers (Hall et al., 2014). In the Northwest (NW) Mediterranean, Ballesteros et al. (2018a) analysed and compared the risk of flooding in the central part of the Catalan coast due to flash floods, storm waves, and sea-level rise; they concluded that flash floods induce higher risks in comparison with marine-related flooding, even though they are acting on a smaller spatial scale along the coastline. However, they did not consider these different drivers to jointly contribute to compound flooding. With respect to compound flooding, most of the existing analyses are part of very large-scale studies (e.g. Bevacqua et al., 2019; Paprotny et al. 2020), with few examples at smaller regional scales (Wahl et al., 2015; Wu et al., 2018; Hendry et al., 2019). Among them, Bevacqua et al. (2019) identified Mediterranean coasts as the European areas with the highest probability of compound flooding under present conditions. As is the case with most existing studies of compound flooding, they considered that storm surge and precipitation were climate drivers that would act simultaneously. When characterising coastal compound flooding from a risk-oriented perspective, the definition of the compound event itself and the choice of contributing climatic drivers are key aspects to be considered. Recently, Zscheischler et al. (2020) proposed a classification of compound events into four main types, which facilitates the analysis of the mechanisms driving the impact and thereby provides a framework for risk adaptation. Using this classification, this study considers and analyses two main types of events: multivariate and spatially compounding.

A *multivariate compounding event* refers to the co-occurrence of hazards from multiple climate drivers in the same geographical region. This is the most common type of event when analysing compound coastal flooding, as defined by the co-occurrence of a marine driver (usually storm surge) and a 'terrestrial' one such as rainfall or river flow acting at the same site (e.g. Wahl et al., 2015; Hendry et al., 2019). Due to the characteristics of coastal storms in the NW Mediterranean, waves are considered the main marine driver controlling the floodwater volume to the hinterland, since the wave-induced run-up, $R_u$, is much larger than the magnitude of the storm surge (e.g. Mendoza and Jiménez, 2009; Mendoza et al., 2011). Moreover, the use of storm waves as the marine driver also potentially indicates the importance of interconnected erosion hazards (in addition to flooding). On the other hand, due to the nature of flooding in the NW Mediterranean coastal zone, heavy rainfall episodes are considered the main terrestrial drivers (as a proxy for runoff), which lead to flash floods (Cortès et al., 2018; Gaume et al., 2019).

*Spatially compounding events* refer to co-occurring hazards from different climate drivers at distant locations within a limited time window. From a risk management standpoint, these events are very relevant because they may overwhelm the capability of emergency-response services since these have to respond to a large number of emergency situations throughout the region at the same time. In this study, these events are defined by the co-occurrence of the two above-mentioned hazards, heavy rainfall and coastal storms, within a time window of three days along the Catalan coast (NW Mediterranean, Spain). This time interval is used in the area to identify independent episodes between consecutive events. Thus, a location under such an event will experience only heavy rainfall or a coastal storm that will accumulate with hazards happening simultaneously, or in rapid succession, in other parts of the territory.

To put this study in the context of risk management, this work will also illustrate the associated impact of selected compound events. In any case, the impact is likely the result of a combination of climatic and societal drivers, with the climate drivers controlling the magnitude of the hazards (analysed herein) and the societal drivers causing an increase or decrease in the associated impacts (e.g. Raymond et al., 2020). One of the problems in properly accounting for these impacts in large geographical areas is the difficulty in obtaining after-event local data across the entire territory. However, a way to identify remarkable events is by considering the significance of their associated impacts in qualitative terms by analysing after-event press coverage and/or insurance data. In the study area, this has been done previously by Llasat et al. (2009) and Cortès et al. (2018) for flash floods and by Jiménez et al. (2012) for coastal storms. This will also be the approach adopted to illustrate the impact of selected events herein.

Within this context, the main aim of this work is to characterise the occurrence of compound flooding events along the Catalan coast (representative of the NW Mediterranean). To this end, we investigated the dependency between coastal storms and intense rainfall for the two types of compound events previously introduced: multivariate events, in which we search for the simultaneous presence (within a time range of three days) of a coastal storm and a heavy rainfall episode in the same geographical area, and spatially compounding events, in which we search for the simultaneous presence (within a time range of three days) of a coastal storm and a heavy rainfall episode in different geographical areas. Thus, (i) we quantify the occurrence frequency of the different types of compound events; (ii) we analyse the spatial variability of the different types of compound hazards and the dependence between extreme variables (rainfall and wave height); and (iii) we examine the prevailing synoptic meteorological patterns during the compound events to identify whether the meteorological drivers can be distinguished in terms of event type (multivariate vs. spatially compounding) and the intensity of the drivers. Finally, some examples of the identified events are outlined in terms of their characteristics and induced impacts.

The remainder of this paper is organised as follows. Section 2 introduces the study area and describes the data used. Section 3 presents the methodology used in the analysis. Section 4 presents the results of analysing compound events along the Catalan coast, and illustrates them with selected remarkable events occurring in the study area during the last 30 years. Section 5 discusses these results. Finally, conclusions are presented in Section 6.

## 2. Study area and data

### 2.1. Study area

The study domain is in the north-east of the Iberian Peninsula and consists of the coastal zone along Catalonia and the river basins flowing into it, which are composed of the internal river basins of Catalonia and the Ebro lower river basin (Figure 1). The coastline runs in the SE–NE direction and is bounded by the presence of two parallel mountain ranges located close to the sea: the littoral range (maximum altitude around 600 metres above sea level, masl) and the pre-littoral range (maximum altitude around 1800 masl). The northern part of the region is limited by the Pyrenees, running from west to east, with altitudes greater than 2000 masl. Therefore, the region is prone to the development of flash floods and thunderstorms (Llasat et al., 2014b), both from a hydrological point of view (existence of many small torrential catchments) and from a meteorological point of view (i.e. orographic forcing of Mediterranean air masses) (Llasat and Puigcerver, 1992). In fact, the impact of mountains on the low-level wind circulation usually triggers convective instability and affects the pressure fields (Jansà et al., 2014).

The Catalan coastline extends about 600 km, of which ~280 km corresponds to sedimentary coasts. The combination of the decrease in river sediment supplies, current level of urbanisation and infrastructure development, and the natural littoral dynamics has led to an overall shoreline erosion during the last few decades (Jiménez and Valdemoro, 2019). From the

perspective of coastal storms, the area is subjected to dominant NE–E extreme waves as well as secondary impacts from the S–SE (Mendoza and Jiménez, 2009; Bolaños et al., 2009; Mendoza et al., 2011). The NW Mediterranean is a microtidal environment with an astronomical tidal range of about 0.25 m. Meteorological tides are of low amplitude, reaching maximum recorded values up to 0.5 m during favourable conditions (under low atmospheric pressure centres and landward-blowing

winds), in such a way that they are much lower than the wave-induced Ru during coastal storms (Mendoza and Jiménez, 2009). The order of magnitude of the storm-induced coastal hazards in the study area can be seen in Mendoza and Jiménez (2009) and Bosom and Jiménez (2011). Although the coastal storm intensity has not significantly changed (e.g. Casas-Prat and Sierra, 2010), the wave action on a progressively narrowing coastline has resulted in a significant increase in coastal damages during the last few decades (Jiménez et al., 2012).


To perform an integrated study of the 'terrestrial' (rainfall) and 'coastal' (waves) compound events, the study region was divided into seven areas following previous studies on flash floods (e.g. Llasat et al., 2016), dividing the region into its main groups of natural catchments along the coast (Figure 1). Each area is composed of several river catchments and/or groups of torrential catchments flowing to their corresponding coastal stretch.

**2.2. Data**

Three main climatic datasets were used in this work to characterise the rainfall, coastal storms, and weather conditions. Rainfall was characterised using daily rainfall (P24h) data obtained from the Spanish Meteorological Agency (AEMET) database (Ramis et al., 2013), which includes records from 491 automatic weather stations (AWS) in Catalonia (Figure 2) covering (non-homogeneously) the period 1950–2015. The selection criteria for identifying records to be used in this analysis consisted

of identifying those AWS belonging to catchments in coastal regions with a homogeneous coverage of the 41-year period from 1973 to 2013, resulting in 69 case-study rain gauges (Figure 2 and Table 1). Flood impacts were obtained from the INUNGAMA database, which contains all the flood events that have affected Catalonia since 1981 as well as all the catastrophic events since 1900 (Barnolas and Llasat, 2007; Llasat et al., 2016).

The wave data used were obtained from the hindcast Downscaled Ocean Waves (DOW) dataset (Camus et al., 2013), which was derived from the Global Ocean Waves dataset (Reguero et al., 2012). Data consisted of hourly values of hindcast wave conditions characterised by the significant wave height (Hs), wave period, and mean wave direction covering the same period as the rainfall data (1973–2013). The datasets were retrieved for 19 nodes located nearshore (about 20 m water depth), homogeneously covering the analysed basins along the coast (Figure 1 and Table 1) to properly capture regional variations in

the storm climates. Additionally, wave records during the Gloria storm in January 2020 were obtained from the SIMAR database from Puertos del Estado (www.puertos.es/es-es/oceanografia).

Weather conditions were characterised by geopotential fields at 1000 hPa from the US National Center for Environmental Protection (NCEP) datasets. NCEP/NCAR Reanalysis I (1948–present) and NCEP/DOE Reanalysis II (1979–present)

generated by the National Oceanic and Atmospheric Administration (NOAA) were used. In the first case, NCEP considered the same climate model that was initialised with different types of weather sources (Kalnay et al., 1996). The second version of the first reanalysis considers the starting point of the major satellite era, which implies that more observations included fewer errors in the resulting fields (Kanamitsu et al., 2002). To retrieve weather data from the NCEP/NCAR Reanalysis datasets, we used the RNCEP library of R-Cran (Kemp et al., 2012). The fields were collected for a spatial extent covering

longitudes −25° to 29° and latitudes 30° to 64°, with a spatial resolution of $2.5° \times 2.5°$ ($N = 13 \times 15$ grid) covering the period 1973–2013 with a temporal resolution of 6 h.

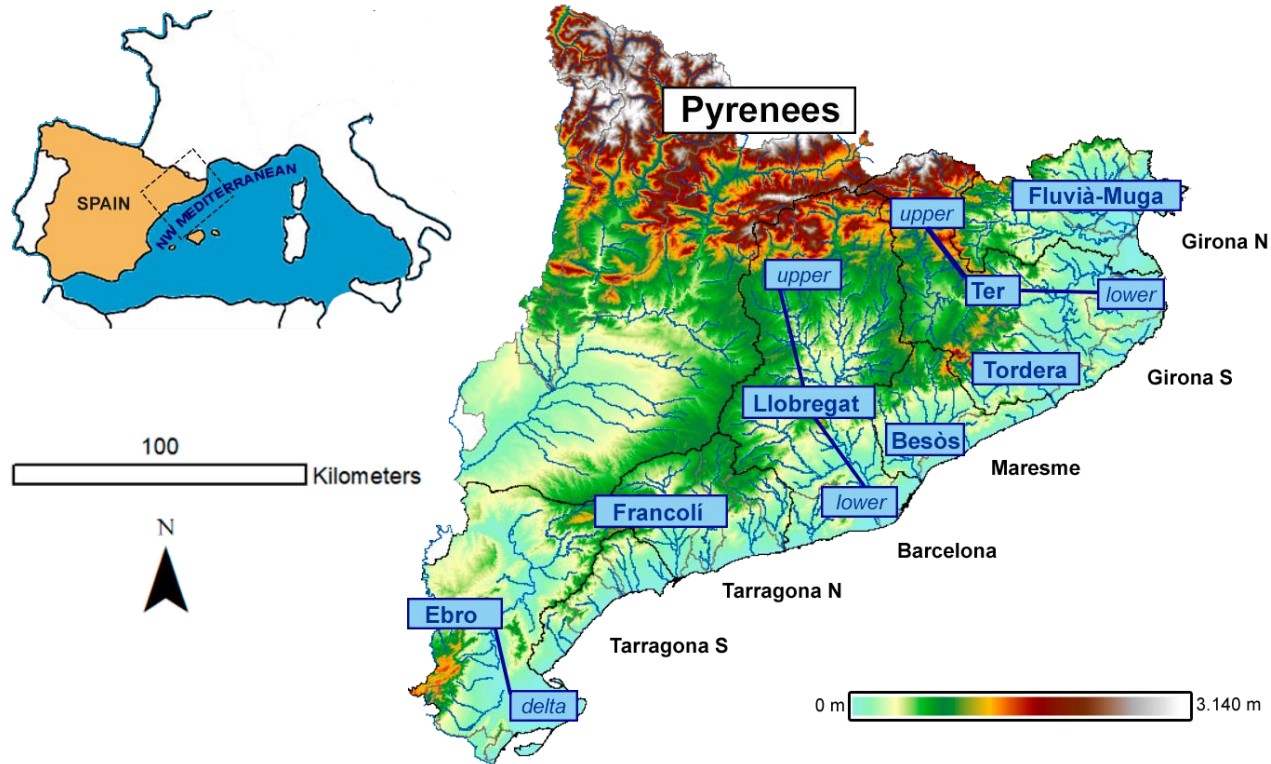

**Figure 1. Study area and main coastal river systems. Digital elevation model 15x15m by Institut Cartogràfic and Geològic de Catalunya (ICGC)**


**Table 1. Number of selected rain gauges and waves nodes in the different areas along the coast (Figure 1).**

| ID | Basin | Rain gauges | Wave nodes |
|---|---|---|---|
| Area 1 | Girona N | 6 | 3 |
| Area 2.a | Lower Ter and Tordera | 9 | 3 |
| Area 2.b | Upper Ter Basin | 9 | - |
| Area 3 | Maresme | 7 | 3 |
| Area 4.a | Lower Llobregat basin | 5 | 3 |
| Area 4.b | Upper Llobregat Basin | 6 | - |
| Area 5 | Tarragona N | 8 | 2 |
| Area 6 | Tarragona S | 5 | 2 |
| Area 7 | Lower Ebro and delta | 14 | 3 |
| **TOTAL** | | **69** | **19** |

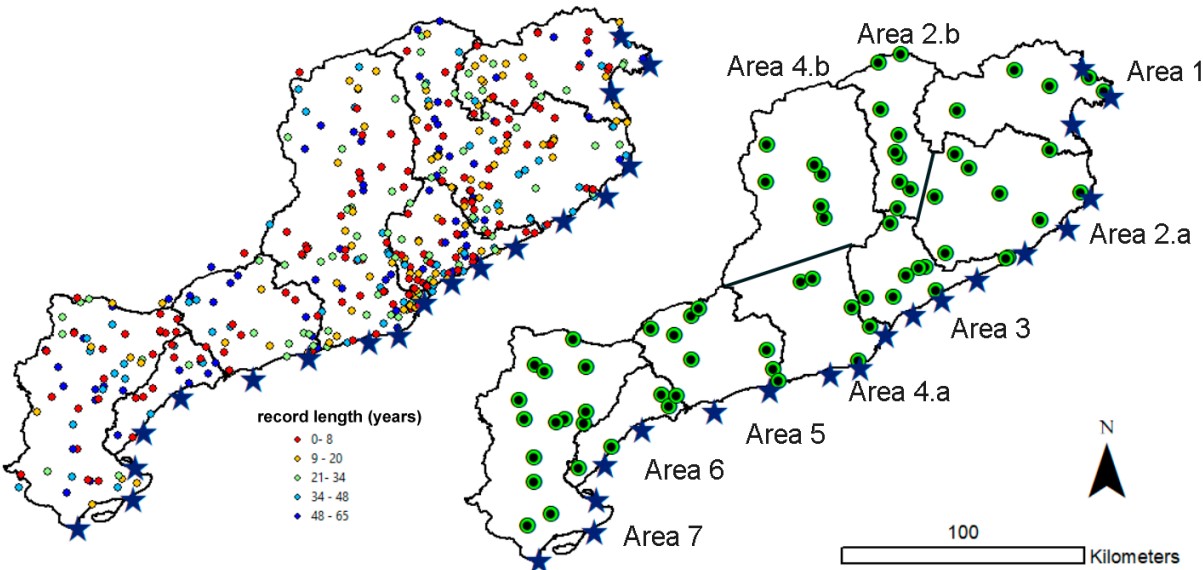

**Figure 2. Location of existing rain gauges, AWS (coloured dots), and wave nodes (stars) in the different drainage basins along the Catalan coast (left). Selected AWS per drainage basin (areas) along the coast (right).**

## 3. Methods

### 3.1. General framework

The general methodological framework adopted in this study consisted of the following steps (Figure 3):

(i) Identification of compound events. First, individual heavy rainfall and coastal storm episodes are identified at all rain gauges and coastal nodes. Then, compound events are defined by identifying dates upon which the two considered drivers co-occur along the territory. Extreme events with only 'pure' coastal storms without rain or 'pure' rain episodes without waves are discarded. Each compound event is characterised in terms of a representative Hs (the maximum value reached during the event) and daily precipitation (P24h) per coastal basin.

(ii) The results from (i) are used to assess the frequency of occurrence and spatial distribution of the different event types (multivariate and spatially compounding). At this stage, the correlation between driver intensity (i.e. the correlation between the maximum Hs and P24h) is also analysed for both event types.

(iii) Compound event dates obtained in (i) are used to retrieve 1000 hPa geopotential maps, which are then classified using correlation-based techniques to determine the main associated weather types. Notably, an event can be associated with different weather types throughout its lifetime.

(iv) The results in (i) and (iii) are combined by feeding a Bayesian network (BN) to characterise each weather type in terms of the spatial distribution of the multivariate and spatially compounding events and the probabilities of exceedance of driver intensities (P24h and Hs).

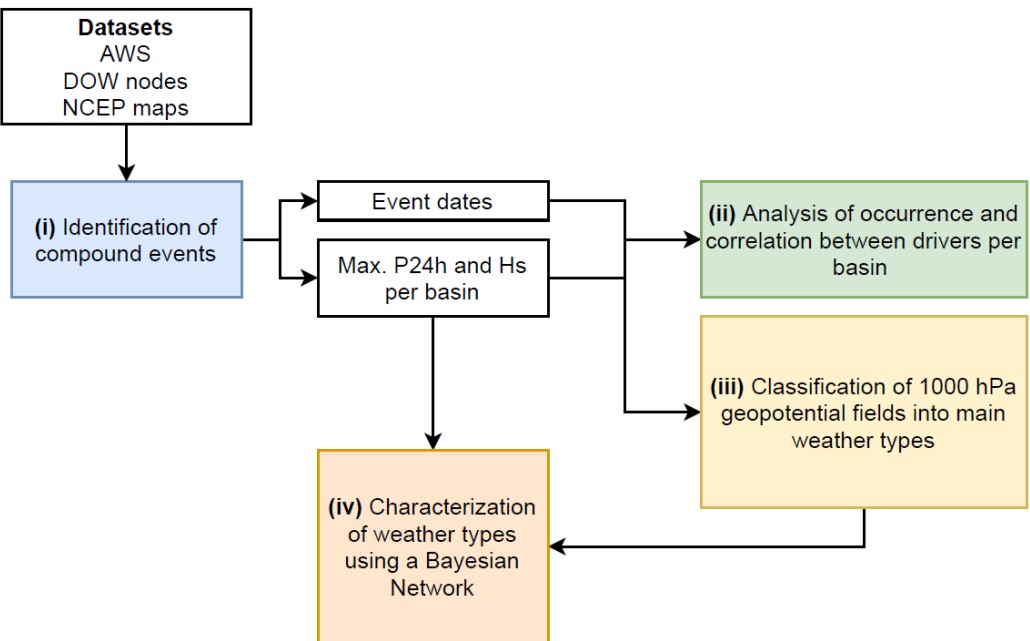

**Figure 3. General methodological framework.**

### 3.2. Identification of extreme events

The first step consisted of identifying individual extreme events from selected rain and wave datasets (Figure 1). This was done by applying the peak over threshold (POT) method to daily rainfall (P24h, in mm) and to significant wave height (Hs) as representative climate variables of the compounding drivers: heavy rainfall and coastal storms.

Following previous studies in the area (Barbería et al., 2014; Cortès et al., 2018), potential extreme rainfall episodes were
identified using a P24h of 40 mm and a three-day interval between consecutive events to identify independent episodes. Although the daily threshold is below the value used by some international projects such as MEDEX (Jansà et al., 2014 proposed 60 mm), local flash floods and urban floods can be produced when this precipitation falls within less than 2 h. The event is then characterised by the maximum recorded P24h value. A second threshold of 100 mm was used to flag severe extreme events to later differentiate them during the characterisation assessments. This threshold has previously been used in
the region when studying extreme rainfall associated with large riverine flooding (Gilabert and Llasat, 2018).

Coastal storms were identified using a double-threshold POT (see Sanuy et al., 2019). The 98[th] percentile of the Hs time series is used as the first filter to locate storm start and end times, which roughly correspond to Hs = 2 m, in agreement with Mendoza et al. (2011) for NW Mediterranean conditions. Then, an upper threshold given by the 99.5[th] percentile is applied to retain only
extreme storms. This criterion results in class III storms according to the Mendoza et al. (2011) classification, which corresponds to the minimum required conditions to produce significant impacts on the coast (i.e. erosion and inundation, see Mendoza and Jiménez, 2009). In addition, a three-day interval of fair-weather conditions between consecutive events was used to identify independent episodes.

The result of this step is a collection of heavy rainfall and wave storm individual episodes for all AWS and wave nodes. Each episode is characterised by an initial and final date and the maximum values of P24h and Hs for rainfall and waves, respectively.

### 3.3. Compound event classification and occurrence

The second step consisted of identifying and characterising compound events in each area along the coast (Figure 2). First, we established the occurrence of a coastal (wave) storm event by comparing the storm initial time at all coastal nodes within a 24-hour window. All storms within such a window were considered a single event, which is labelled with the same date, and they should correspond to a coastal storm propagating along the territory. Then, all rain gauges were surveyed to identify the occurrence of heavy rainfall episodes during the identified coastal (wave) storm and in the three days before. If no heavy rainfall is registered at any rain gauge, the event is removed from the analysis, as it would correspond to a pure wave storm. On the contrary, if any station registers an extreme P24h episode, the event is flagged as a compound event, with its date of occurrence being given by the earliest starting time of coastal storms at any node. Finally, each compound event is characterised by the maximum P24h and Hs values at the stations and nodes within each coastal area, and during the event duration.

As a result of this process, a compound event is identified herein when a heavy rainfall episode at any station along the coast occurs simultaneously, or in rapid succession (within a three-day interval), with extreme waves at any location along the coast. Therefore, an identified compound event at a given time may present different characteristics along the costal basins (hereafter also named areas or sectors): areas in which rainfall and wave extreme events simultaneously co-occur, areas with only one extreme component (either rain or waves), and areas without any extreme episodes. Then, each compound event is characterised along the coast (in each sector) as follows: (i) multivariate (simultaneous rainfall and wave episodes); (ii) spatially compounding (SC) rain, where local extreme conditions correspond to rainfall; and (iii) spatially compounding (SC) waves, where local extreme conditions correspond to storm waves. The classification intends to classify the event as it is experienced in each basin. Thus, in the face of a compound event (regional scale) there will be basins that experience it as multivariate (both components co-occur) and basins that experience it as spatially compounding. In the second case, the basin may be receiving only rain (SC-rain) or only waves (SC-waves). According to our definition of a compound event, there will always be a co-occurrence of the two components at the regional scale

### 3.4. Intensity correlation analysis

To investigate the correlation between the magnitude of both components of the compound event, P24h and Hs, across all areas, we used the Spearman rho coefficient (e.g. Genest and Favre, 2007); this is defined as the Pearson correlation coefficient between the variable ranks (eq. 1). The correlation of the wave magnitude (Hs) at each coastal area with the P24h rainfall at each of the nine areas is calculated as follows:

$$\rho = \frac{cov(rg_{Hs}, rg_{P24h})}{\sigma_{rg_{Hs}} \sigma_{rg_{P24h}}} \tag{1}$$

where $rg_{Hs}$ and $rg_{P24h}$ are the ranks of Hs and P24h, respectively, $cov$ is the covariance, and $\sigma$ is the standard deviation.

### 3.5. Synoptic typology

During the development of a compound event, different weather types can be present; in this study, we therefore use corresponding pressure fields (geopotential height at 1000 hPa) at the closest time to the date-time assigned to each event (Section 3.3). Weather conditions were then classified by applying a correlation-based method (Yarnal, 1993; Wu et al., 2018) to the 1000 hPa geopotential height field. The method consists in obtaining map patterns using the Pearson product-momentum correlation (rxy, eq.2) to depict the degree of similarity of spatial structures between pairs of gridded data (i.e. the map typing focuses on the positions of high- and low-pressure centres, rather than their magnitudes). All maps were extracted at the closest time to the beginning of the compound event, as defined in Section 3.4 (i.e. the beginning of the coastal storm).

First, all maps are normalised via $Zi = (zi - \bar{z})/\sigma_z$, where $Zi$ represents the number of positive or negative standard deviations from the mean at each grid cell $i$, $z_i$ is the original value at grid point $i$, and $\bar{z}$ and $\sigma_z$ are the mean and standard deviation of the $N = 13 \times 15$ grid point values. Once normalised, each map is compared with all other maps using:

$$r_{xy} = \frac{\sum_{i=1}^{N}[(x_i - \bar{x})(y_i - \bar{y})]}{\sqrt{\sum_{i=1}^{N}(x_i - \bar{x})^2(y_i - \bar{y})^2}} \qquad (2)$$

where $x_i$ and $y_i$ represent the normalised value at each of the $N$ points of the pair of maps being compared, and $\bar{x}$ and $\bar{y}$ are the corresponding means across the $N$-point grids. A pair of maps is considered similar when $r_{xy} \geq rt$, where $rt$ is a correlation threshold. Different sources of subjectivity exist in choosing the value of $rt$ (Yarnal et al., 1993), which usually depends on a balance between the number of identified patterns and the number of dates (events) remaining without classification.

The process was applied as described by Wu et al. (2018). The first date of the reference is used as the key day, and all maps are compared to create the first class. Then, all classified dates are removed, and another date from the non-classified pool is used as the second key day. A value of $rt = 0.2$ was used in this step, which led to four weather-type candidates. Then, a second comparison between each individual map and the average of each group candidate was performed to ensure that maps that are similar to more than one type are classified into the group corresponding to a maximum $r_{xy}$. This led to a final three-group classification with a mean $r_{xy}$ of 0.64–0.7 per group.

## 3.6. BN-based classification

BNs are statistical tools based on acyclic graph theory and Bayes theorem (Pearl, 1988; Jensen, 1996) and have demonstrated their versatility and utility in efficiently combining multiple variables to predict or characterise system behaviour (e.g. Gutierrez et al., 2011; Plant et al., 2016; Beuzen et al., 2018). The BN is used here to assess the probabilistic relationship between each type of meteorological forcing (Section 3.5) and their associated effects (multivariate or spatially compounding) and driver intensities (Hs and P24h) at each of the different basins.

Figure 4 shows the BN structure, the considered variables, and the variable discretisation. The arrows depict the parent–child relationships, i.e. the results will be presented in terms of the probability of given values of Hs, P24h, and impact type conditioned to the different possible combinations of the synoptic case and area. The training dataset consisted of 1260 variable combinations resulting from the previous assessment of 140 compound events in nine different areas.

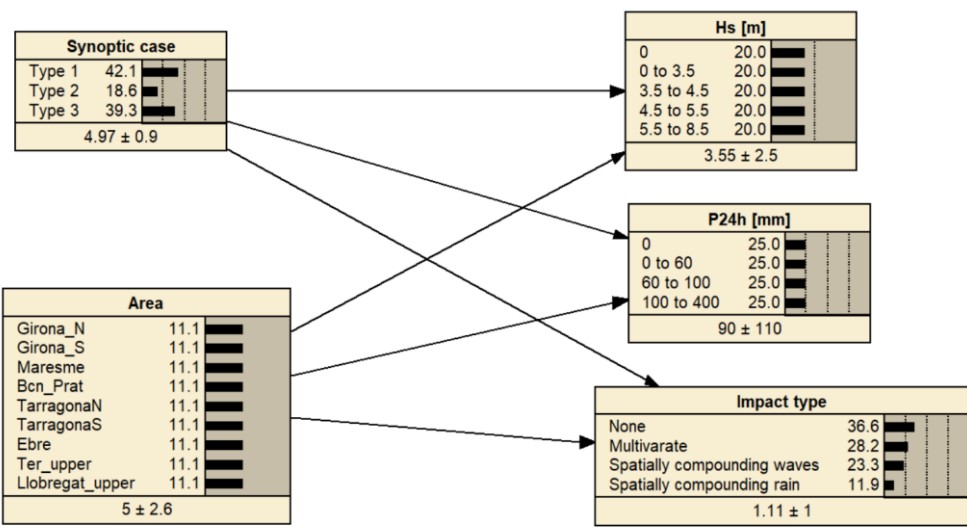

 **Figure 4. BN configuration used to characterise the system behaviour. The severity of the compounding forcing (Hs and P24h) and the presence of multivariate or spatially compounding (wave or rain only) effects are conditioned to the synoptic case (weather type) and area (basin/coastal sector).**

## 4. Results

### 4.1. Frequency and location of compound events

During the analysed 1973–2013 period, 225 coastal storms and 605 heavy rainfall episodes affecting at least one of the considered areas were identified. It must be considered that the real number of rainfall events would be higher, since convective localised episodes are under-represented in the used dataset. From this total, 140 episodes can be classified as compound events in which wave storms are accompanied by heavy rainfall in any area along the coast. This means that 62% of coastal storms and 23% of heavy rainfall episodes can be labelled as compound events and that the probability of having such compounding conditions is larger under coastal (wave) storms. The average frequency of occurrence along the Catalan coast during the study period was 3.4 compound events per year, without presenting any statistically significant trend during the analysed 41 years (Figure 5).

Figure 6 and Table 2 show the spatial distribution of the 140 identified compound events along the Catalan coast according to their typology as locally recorded: multivariate (simultaneous rainfall and wave storm episodes in the same area), SC-waves (a solo wave storm episode in a given area with simultaneous heavy rainfall co-occurring at a different area), and SC-rain (a solo rainfall episode in a given area with a simultaneous wave storm co-occurring at a different area). In addition, the local absence of extreme conditions for both drivers was retained.

The results show that, in the presence of a compound event along the Catalan coast, areas with the highest probability of experiencing a multivariate event are in the northernmost part, Girona N, and the Lower Ter–Tordera basins (with an occurrence frequency of about 55%) followed by the southernmost end in the lower Ebro basin and delta (with an occurrence frequency of about 40%). On the other hand, although the areas located at the central part of the coast show a non-negligible probability (about 20–30% of recorded events) of experiencing multivariate events, they are dominated by the presence of spatially compound events with the local presence of wave storms (about 40–50% of recorded events). These results would indicate that, in the study area, when a regional compound event occurs, wave storms are the 'spatially dominant' driver, with all areas along the coast having a probability greater than 60% of having local wave storms (either multivariate or SC-waves). On the other hand, areas presenting a high probability (>60%) of having rainfall extremes (either multivariate or SC-rain) during regional compound events are restricted to the two northernmost areas and the southernmost one (58%); the central part of the coast (Areas 3 to 6) presents relatively low probabilities of experiencing extreme rainfall (35–42%). All given percentages are relative to the total number of identified compound events.

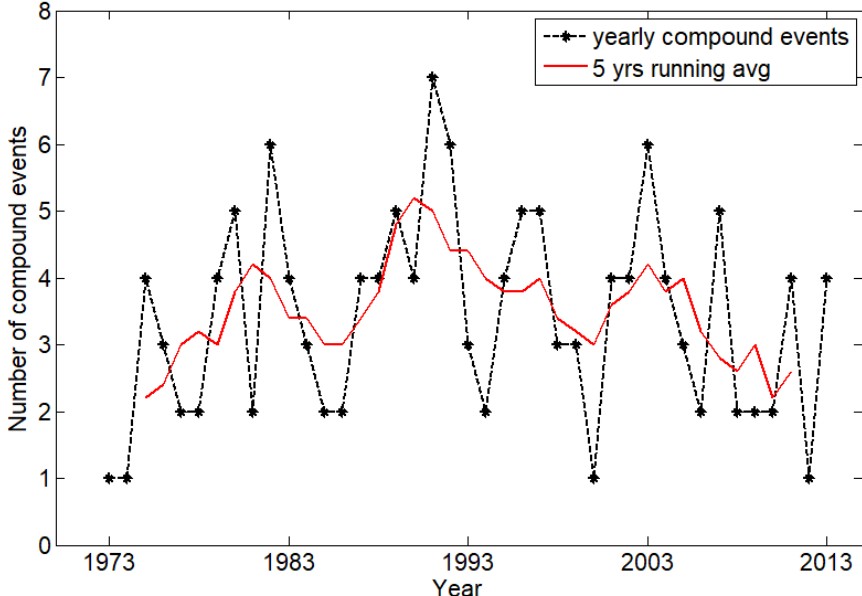

**Figure 5. Annual number of compound events (black) and the running five-year average (red) for the period 1973–2013.**

Rainfall events in the 'terrestrial' areas which correspond to the Ter and Llobregat upper basins (areas 2b and 4b) are filtered with P24h >100 mm to assess their potential effects at the coastal fringe; fewer than 5% of cases reach those precipitation levels in combination with extreme waves at the coast (table 2).

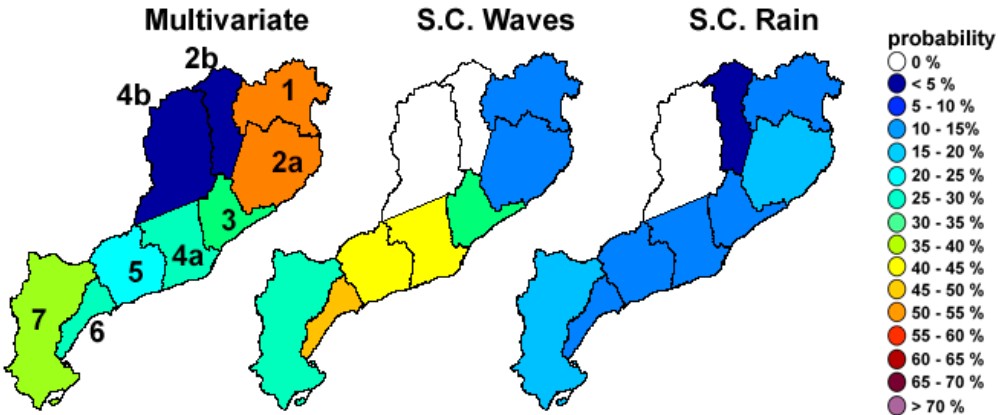

**Figure 6. Probability of occurrence of the different types of compound events along the Catalan coast. Multivariate (local simultaneous rainfall and wave storm episodes); SC-waves (local wave storm episodes and simultaneous rainfall in a different area); SC-rain (local rainfall episode and simultaneous wave storm in a different area). Probabilities are given with respect to the presence of a compound event (average occurrence during the period 1973–2013 of about 3.4 events per year). Probabilities are given per each area along the coast, and the sum of the different types of events per area does not necessarily reach 100% due to cases in which**
**neither rainfall nor wave storms locally occur. Area numbers are specified in left map.**

**Table 2. Number of compound events per type of occurrence of climatic drivers at each basin. Upper basins (not directly affected by waves) are marked as (*) where a threshold of P24h > 100 mm is used.**

| ID (Figure 1) | Basin | Multivariate | SC-waves | SC-Rain | No driver |
|---|---|---|---|---|---|
| Area 1 | Girona N | 70 | 16 | 19 | 35 |
| Area 2.a | Lower Ter and Tordera | 76 | 14 | 28 | 22 |
| Area 2.b | Upper Ter Basin | 3(*) | - | 5(*) | 132 (*) |
| Area 3 | Maresme | 43 | 48 | 17 | 32 |
| Area 4.a | Lower Llobregat basin | 36 | 58 | 18 | 28 |

| Area 4.b | Upper Llobregat Basin | 4 (*) | - | 0 (*) | 136 (*) |
|---|---|---|---|---|---|
| Area 5 | Tarragona N | 34 | 57 | 20 | 29 |
| Area 6 | Tarragona S | 35 | 65 | 15 | 25 |
| Area 7 | Lower Ebro and delta | 54 | 36 | 28 | 22 |

**4.2. The correlation between wave components and rainfall intensity during compound events**

Once the probability of occurrence of compound event was analysed at different areas along the coast, the correlations among the magnitude of the climatic drivers (rainfall and waves) needed to be determined. Figure 7 shows the computed Spearman rho coefficient by correlating the waves (Hs) in a given area (marked with a diamond in the figure) with rainfall (P24) at all areas along the coast.

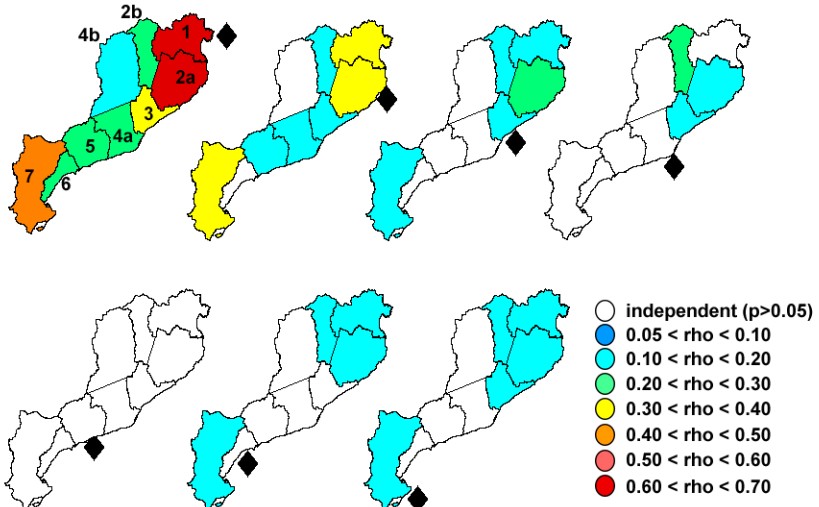

**Figure 7. Correlation values (Spearman rho) between the Hs magnitude and P24h during compound events. Each map shows the correlation between the waves (Hs) at the area indicated by the diamond and rainfall in the other areas. White areas indicate that variables are statistically independent at a significance level of 0.05. Area numbers are specified in top left map.**

The results show that, in general, the correlation between the intensity of local wave storms and rainfall across the territory (measured as peak values of Hs and P24h during the event) decreases from north to south, following the observed trend in the dominance of multivariate events. The highest correlation value (rho = ~0.65) was obtained for multivariate events in the northernmost area (Figure 7), suggesting a strong link between locally simultaneous wave storms and rainfall. The connection between these drivers extends southwards in such a way that the correlation between the wave storm intensity in Area 1 with simultaneous rainfall episodes in the adjacent Area 2 is of the same order of magnitude. The correlation progressively decreases southward as we compare it with the rainfall recorded in the central basins, but, in all cases, the correlation is statistically different from zero. Notably, a value of rho > 0.4 is obtained in the southernmost sector, where the presence of multivariate events is higher than in the central basins.

When the intensity of wave storms recorded in Area 2 is correlated with rainfall across the territory during compound events (Figure 7), a similar behaviour than that in Area 1 is observed, although with lower correlation values. As we progressively move to the south, the correlation between the intensity of the local wave storms and P24h at any area consistently decreases to very low values or, directly, they are statistically uncorrelated. On a regional scale, the central basins have the lowest values of rho, suggesting an independence of storm waves and intense rainfall events.

### 4.3. Synoptic conditions

Weather conditions during the 140 identified compound events were classified in three different synoptic types (Figure 8). Synoptic Type 1 conditions prevail during 42.1% of cases and are characterised by the presence of lower pressures northwest of the Iberian Peninsula over the Atlantic Sea and higher pressures in the Central Mediterranean. The deep low in the north-western part of the Iberian Peninsula and the strong anticyclone over centre Europe favour a strong pressure gradient and consequently induce intense winds from the south (Llasat, 1987). This type of situation usually creates a mesoscale structure when they impinge over the Pyrenees Range, known as an orographic dipole, with a mesoscale high over Catalonia that modifies the synoptic pressure field and creates an eastern component of the wind that favours the entrance of warm and wet air. At the same time, the mountain range triggers potential instability and develops convective systems and heavy rainfall (Trapero et al., 2013; Llasat et al., 2014b).

Synoptic Type 2 (18.6% of cases) is characterised by the presence of a depression in the south-eastern Iberian Peninsula and an anticyclone in the northwest, which creates a strong pressure gradient and N-NE winds. This kind of meteorological situation is more effective in generating sea storms than it is in generating heavy rainfall. Synoptic Type 3 conditions (39.3% of cases) are similar to those of Type 2, with a deep low in the southern Iberian Peninsula and an anticyclone to the north-east. The main difference is that this anticyclone is placed over the centre of Europe in this type; this pattern gives rise to a strong E-SE wind at low levels.

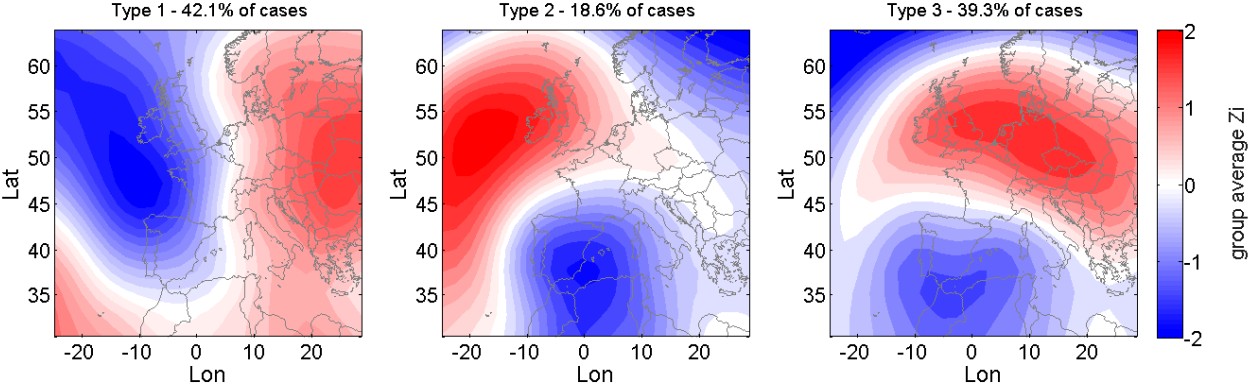

**Figure 8. Synoptic types during the occurrence of compound extreme events along the Catalan coast based on the correlation between the 1000 hPa geopotential fields (coloured shades). The group average Zi represents the mean number of positive or negative standard deviations from the mean at each grid cell *i*.**

### 4.4. Compound event characteristics under each synoptic type

The BN was used to calculate the probability of occurrence of multivariate and spatially compounding (wave or rainfall) events in different areas given the different synoptic types (Figure 9). The BN was also used to calculate the probability of exceedance of significant thresholds of Hs and P24h for each type of compound event at the different areas along the coast to assess the intensity of each contributing component (Figure 10).

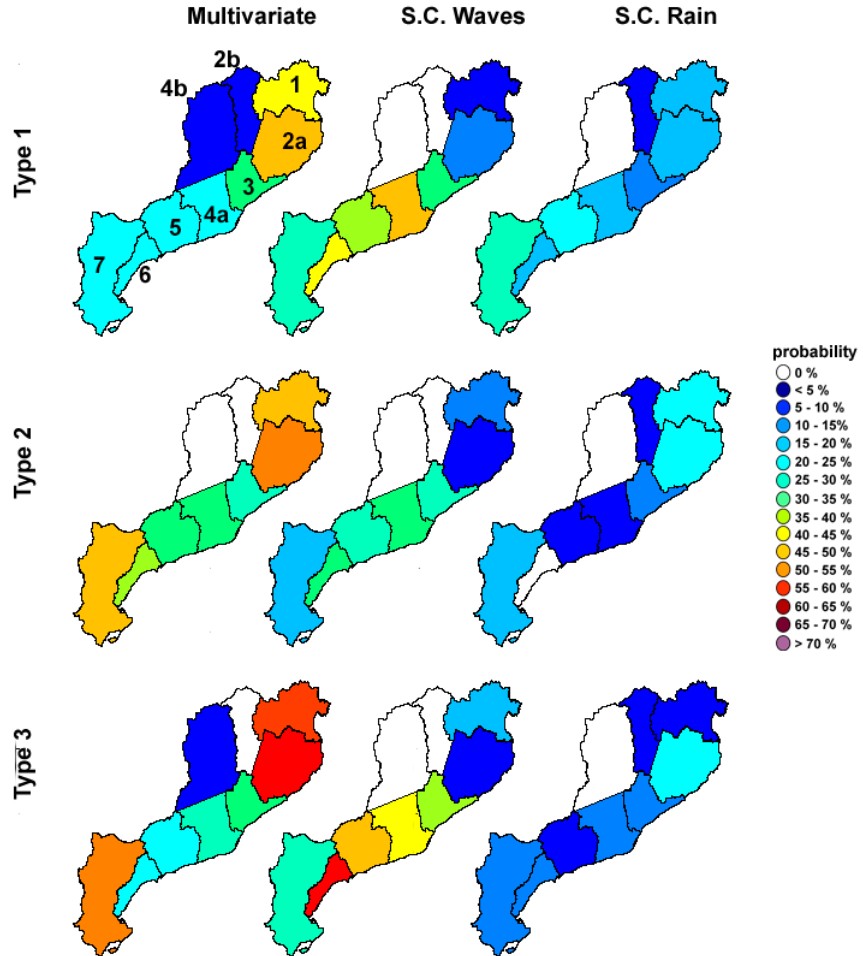

**Figure 9. Spatial distribution of the probability of occurrence of the different types of compound events conditioned to each synoptic type. Probabilities are given per each area along the coast, and when adding the different types of events per area they do not necessarily reach 100% due to cases in which neither rainfall nor wave storms locally occur. Area numbers are specified in top left map.**

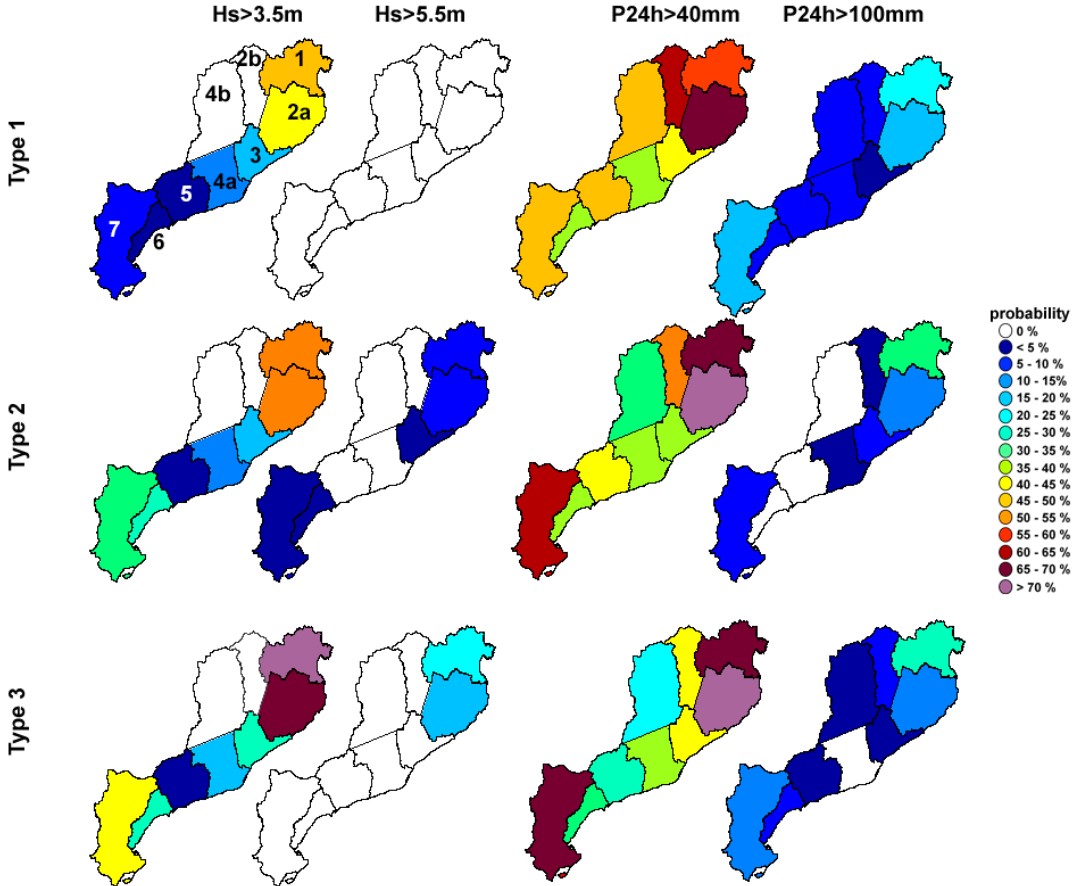

**Figure 10. Spatial distribution of the probability of exceedance for different thresholds of Hs (3.5 m and 5.5 m) and P24h (40 and 100 mm) conditioned to each synoptic type. Area numbers are specified in top left map.**

Under a meteorological forcing generated by a synoptic Type 1, which is the most likely to occur, the probability of occurrence of multivariate events anywhere in the territory is the lowest compared to that associated with the other types (Fig. 9). Moreover, when multivariate events occur, they are concentrated in the northernmost part of the coast (areas 1 and 2). In the rest of the coast, the dominant event is SC-waves. Generated wave storms present the smallest Hs values along the territory, without inducing extreme storms (Hs > 5.5 m) (Figure 10). On the other hand, the probability of SC-rain events is higher than in the other types, especially in the Ebro basin. Moreover, it is the most likely to exceed a P24h of 100 mm.

Synoptic Type 2 episodes are more prone to create Levante (winds blowing from the East) situations, and during which the probability of occurrence of multivariate events increases, especially in the northern and southern areas, while in the central part of the coast multivariate and SC-waves are equally probable (Figure 9). The severity of coastal storms (Hs) increases, especially at the northern and southern ends, where extreme storms (Hs > 5.5 m) have been recorded (Figure 10). The probability of rainfall episodes with P24h > 40 mm increases in the same areas with respect to Type 1, whereas the frequency of the most intense episodes (P24h > 100 mm) decreases, except for Area 1 (Figure 10).

Synoptic Type 3 is similar to Type 2 in that it also represents Levante situations, but is characterized by the marked south-north pressure gradient, which results in significant windstorms and strong waves. Consequently, they present similar overall probabilities of occurrence of multivariate events along the coast, with Type 3 presenting a larger probability of multivariate occurrence at the northern and southern extremes and a higher frequency of SC-waves, especially in the central areas (Figure 9). In terms of intensity, wave storms recorded under synoptic Type 3 present the highest probability of exceeding Hs > 3.5 m in all areas and the highest probability of extreme waves (Hs > 5.5 m), which is restricted to the two northernmost areas (Figure

10). On the other hand, the distribution of probability of P24h exceeding 40 mm or 100 mm is very similar to type 2, since in both cases there is an incidence of a warm and humid air mass from the East.

## 4.5. Compound event characteristics based on historical events

Differences in weather patterns result in events with different characteristics and, consequently, impacts throughout the territory. To put the potential consequences of these events in the context of risk management, the impact of selected events recorded in the study area under the different synoptic types are illustrated with information gathered from after-event press coverage and the INUNGAMA and PRESSGAMA databases (Llasat et al., 2009, 2014a; Jiménez et al., 2012). Synoptic conditions during each analysed event are shown in Figures 11 and 12, with maps being extracted following the criteria described in the methodological framework. Table 3 shows the maximum Hs and P24h recorded in each area along the coast during each event. Two representative events were chosen for Types 1 and 3, as they occur twice as frequently as Type 2 events do, that is only represented here by one event.

Between 6 and 8 November 1982, a compound event generated under a Type 1 synoptic situation (Figure 11) took place along the Catalan coast. From a meteorological point of view, the event was dynamically forced, as it unfolded in the prefrontal and frontal zones of a strong Atlantic baroclinic storm, although the Pyrenees played a relevant role by triggering deep convection. The largest contribution of humidity was from the Atlantic (mainly tropical and subtropical regions but also from the north) with relevant additional input from the western Mediterranean (Insua-Costa et al., 2019). This was a very extensive episode of heavy rain affecting Portugal, Spain, Andorra, and France. Catastrophic flash floods and landslides occurred in the Upper Llobregat Basin (Area 4.b) and Upper Ter Basin (Area 2.b) (Puigdefàbregas, 1983; Llasat, 1987), where nearly 342 mm and 556 mm were recorded in less than 24 h and 72 h, respectively. The Llobregat River (Areas 4.a and 4.b) recorded a peak flow of 1600 $m^3$/s near its mouth when its average discharge was 328 $m^3$/s.

The main peak rainfall and wave conditions recorded during the event along the Catalan coast are shown in Table 3. As can be seen, although waves exceeded storm threshold conditions along the entire coast, their values were relatively low, with only the northernmost sector presenting severe storm conditions according to the Mendoza et al. (2011) classification. Due to this, coastal storm–induced damages were relatively low and were limited to some stretches at Costa Brava (Areas 1 and 2.a) where waves induced minor damage to some marina facilities and caused overtopping at some beach waterfronts. Some beaches in the Maresme region (Area 3) were also affected, with extensive erosion and overtopped promenades. On the other hand, the rainfall-induced damage was extensive and very important, with 14 casualties and 1,033 million € (adjusted for 2020) of private flood damages paid by the Insurance Compensation Consortium (the Spanish public re-insurance company, CCS) as a consequence of the floods in Catalonia (throughout the entire territory and not only in coastal areas). In summary, although this was a multivariate event at many basins, the most important and relevant damage was caused by rainfall-induced floods (Figure 12).

Another significant Type 1 event occurred in 2011, starting on 2 November and lasting until 7 November 2011 (Figure 11). It mainly affected Catalonia (Spain) and Liguria (Italy). In the first region, the maximum cumulative rainfall was 326 mm (close to Area 2.b, Table 3), while the maximum in 24 h was 203 mm (close to Area 4.b, Table 3). It produced a flood in the Muga River (Area 1) with a peak discharge of 378 $m^3$/s near the mouth (on 1 November the flow was 0.7 $m^3$/s) (Llasat et al., 2014). Between 2 and 8 November, the CCS paid 2.1 million € (adjusted for 2020) for damages produced by the sea storm and 458.8 million € (adjusted for 2020) for damages produced by floods in insured assets. Meteorological features showed the presence of a trough at 500 hPa associated with a synoptic frontal wave that evolved into a mesoscale depression along the Catalan coast

on 6 November. This situation favoured the entrance of very warm and wet air from the south-east over Catalonia and humidity advection from the Atlantic.

It should be noted that both events presented similar characteristics, with the most important contribution to damage being induced by rainfall. On the other hand, combined/compound effects were scarcely reported in just few areas, where they locally induce a moderate-to-high impact.

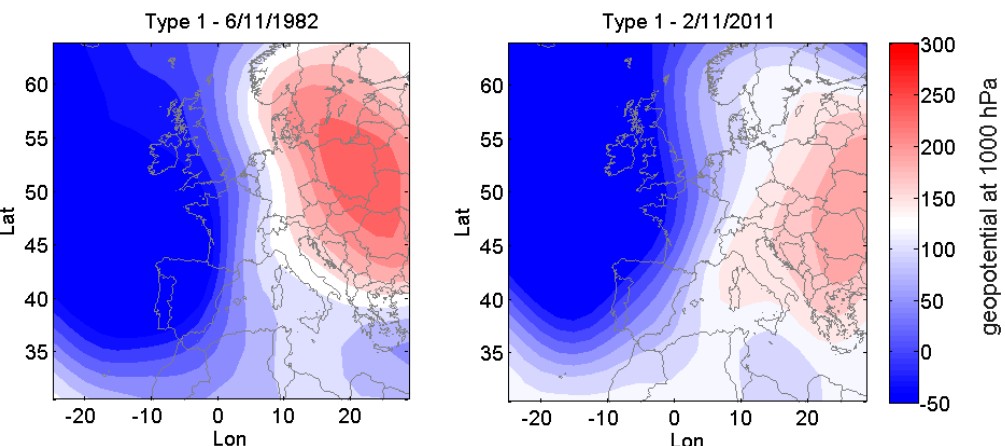

**Figure 11. Maps of the 1000 hPa geopotential fields of Type 1 events extracted at the time closest to the start of the coastal storm.**

An historical compound event generated under Type 2 conditions in the area occurred in November 2001 (Figure 12), when a thermal orographic low over the African plateau interacted with an upper-level trough and developed a strong cyclone that moved toward the Mediterranean Sea over Algiers following a northward trajectory, creating torrential rainfalls and floods in Algiers (more than 700 deaths). When the cyclone reached the sea, the low reached its mature state (Fita et al., 2006; Genovés and Jansà, 2002). The depression was observed at all atmospheric levels, and there was also a zone of high pressure that was located to the north-west of the Peninsula with a pressure higher than 1035 hPa that contributed to the strong wind and pressure gradients. The deep low continued its trajectory to the NE and affected Catalonia, giving rise to a windstorm with a recorded maximum wind speed of 170 km/h (Port-Bou, Area 1). In contrast to the previously described Type 1 situation, this event was characterised by very high Hs values along the entire coast (Table 3), exceeding the threshold for severe storms according the Mendoza et al. (2011) storm classification in nearly all areas (except Area 5) and for extreme storms in the northernmost areas (1 and 2.a). Thus, compounding conditions were wave-dominated, with the rainfall being moderate (although exceeding the 40 mm threshold for P24h) except in the northernmost Area 1, where the P24h was higher than 100 mm. In addition, snowfall in the northern part of the region and severe weather (a tornado in Montgat, Area 3, and hailstorms in Tarragona N and S, Areas 5 and 6, respectively) were also observed. Although many civil protection interventions due to floods and wind action occurred mostly in the central part of the study area, most of the incurred damage was due to coastal (wave) storm-induced hazards. Thus, the entire coastal zone was severely affected from south to north as the storm propagated along the coast (Figure 13). In the southernmost part (Area 7), the Ebro delta plain was extensively flooded, while the beaches were severely eroded. This resulted in significant damage to rice fields and existing infrastructure. Along the entire coast, many ports and marinas were significantly overtopped, with the breakwater of the Port of Barcelona being damaged. In some stretches, such as Barcelona (Area 4.a) and some parts of Maresme (Area 3), many beaches were fully eroded, with their promenades being directly exposed to wave action; the coastal railway along Maresme was also impacted. In the northernmost part of the study site (Areas 1 and 2.a), coastal flooding occurred in several municipalities due to massive overtopping of beaches and

promenades. In summary, although this was a compound event, the most important and relevant damages were caused by coastal (wave) storm-induced hazards (Figure 12).

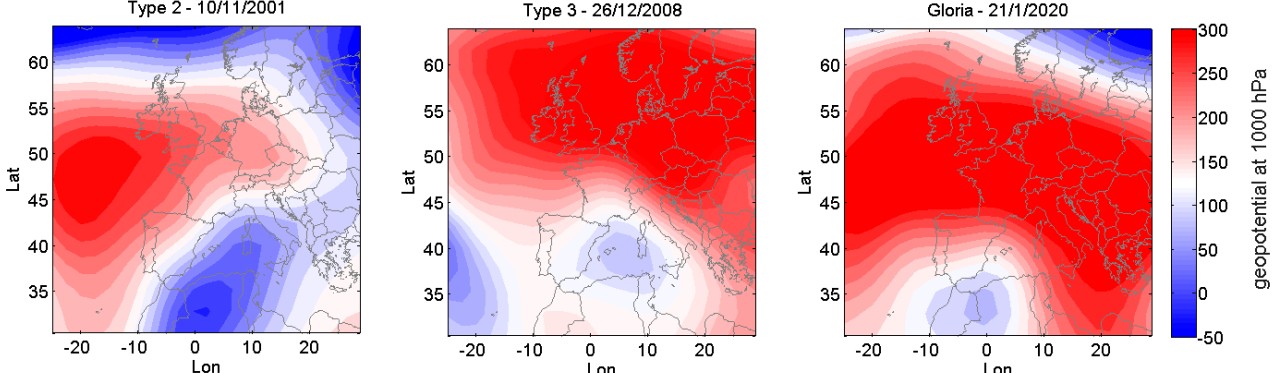

**Figure 12. Maps of the 1000 hPa geopotential fields (shades) of Type 2 and Type 3 events extracted at the time closest to the start of the coastal storm.**

On 26 and 27 December 2008 (Type 3, Figure 12), a very intense coastal storm affected the Catalan coast and was accompanied by strong winds, snow, and rain. The surface synoptic situation by 26 December was characterised by a pronounced anticyclone in northern Europe extending from Ireland to Russia, centred in Denmark, and a low-pressure area with two clear centres over Catalonia and Valencia and another located over the Azores. Along the Catalan coast, there was strong wet advection from the

south-east due to the strong low–high dipole. This situation caused a very high pressure gradient that produced strong advection from the east and south-east of Catalonia. As a result, very high waves and rainfall were recorded along the study area, with the highest values being reached in the northern half of the coast (Areas 1, 2.a, 3, and 4.a). In addition to tangible damage, four fatalities occurred during the episode, three of which were associated with wave action and the fourth occurring from a flood in the Muga River (Area 1 and Table 3). Extreme wave impacts along the coastline induced significant damage, with extensive

sediment losses in the beaches, promenades overtopped, and damage to infrastructure (Figure 13). This occurred especially in the northern part of the coast (Areas 1 and 2.a) where waves reached values typical of extreme storms according to the Mendoza et al. (2011) classification. This was one of the most important recorded coastal storms, with observed impacts also on nearshore ecosystems (e.g. Sánchez-Vidal et al., 2012). On land, the strong wind uprooted a large number of trees and cut off the electricity and telephone lines. The Fabra observatory in the city of Barcelona recorded a maximum wind gust of

approximately 85 km/h. During the event, notable snowfall at low altitudes, some landslides, and a tornado in Platja d'Aro (Area 2.a) were produced. Many roads and train lines were cut off due to heavy snowfall and flooding of the tracks near the sea. The Ministry of the Environment allocated an equivalent of 21.6 million € (adjusted for 2020) to different municipalities on the Catalan coast to carry out emergency work with the aim of repairing the damage caused by the waves on beaches and coastal infrastructure. The government granted aid packages to several municipalities and fishermen for more than 0.72 million

€ (adjusted for 2020).

Recently, the severe storm Gloria took place in the Catalan Sea in January 2020 (Figure 12) with record-breaking events occurring in all areas for both wave heights and rainfall (Table 3). It started as a small superficial depression (about 600 km in diameter) located in the central part of the North Atlantic Ocean, which was increasing  while moving eastward. By 18 January,

the storm had nearly doubled in size, while the Azores High was being reinforced further south. On 19 and 20 January, the high-pressure zone moved to the north of the Gloria storm, leaving the latter over the south of the Iberian Peninsula. Between 20 and 23 January, both areas were well defined in the form of a dipole creating a strong pressure gradient that gave rise to

very intense winds while favouring the entry of maritime air over the study area (Berdalet et al., 2020). The synoptic pattern (on 20 January +00 UTC, closest to the coastal storm start time) corresponded to a Type 3 synoptic event. Wave heights recorded during the peak of the storm along the Catalan coast reached record maximum values, and they were accompanied by the presence of a moderate storm surge, reaching values of ~0.5 m in the southernmost area (Amores et al., 2020; Jiménez, 2020). Wave impacts produced significant erosion at the beaches, with massive overtopping and flooding of low-lying areas such as the Ebro delta, waterfronts, and marinas as well as structural damage in some coastal groins and port breakwaters (Jiménez, 2020). The extreme coastal storm was accompanied by very intense rainfall and thunderstorms throughout the territory, reaching record values from the last seven decades, which significantly contributed to flooding along some coastal plains, occasional cut offs and damage to roads and railways. The Department of Interior of the Government of Catalonia responsible for civil protection services activated three emergency plans for risk management: INUNCAT (flash floods, river floods, and coastal floods), NEUCAT (snowfalls, significant above 600 m), and VENTCAT (wind, extreme gusts lasting about 48 h, with a maximum of 144 km/h). These hazards caused four casualties (and 10 more in the Balearic Islands and Valencia) and extensive damage throughout the territory (Figure 13), with a preliminary evaluation of payments to be covered by the Spanish public re-insurance company (CCS) rising up to about 51 million €. Inversions to rebuild port infrastructures affected by the storm are estimated to be about 17.4 million €; about 6 million € were budgeted by MITECO to repair damage in the public coastal domain, and damage due to floods in the river margins and flood plains was estimated at 42 million € by the Catalan Water Agency (ACA).

**Table 3. Values of the maximum Hs (m) and P24h (mm) for each selected event (Figures 11 and 12) along the study area extracted from the analysis dataset. Data on Gloria (outside the dataset) was extracted from the SIMAR wave database (Puertos del Estado) and XEMA rain gauge system (SMC). Note that the Hs values given for Gloria do not belong to the same database as the other storms do; consequently, their values are not absolutely equivalent.**

| | Type 1 | | | | Type 2 | | Type 3 | | | |
| | 06/11/1982 | | 02/11/2011 | | 10/11/2001 | | 26/12/2008 | | 21/01/2020 (Gloria) | |
| | Hs (m) | P24h (mm) | Hs (m) | P24h (mm) | Hs (m) | P24h (mm) | Hs (m) | P24h (mm) | Hs (m) | P24h (mm) |
|---|---|---|---|---|---|---|---|---|---|---|
| Area 1 | **5.4** | 78 | - | 93 | **7.9** | **107** | **8.0** | 203 | **7.2** | 101 |
| Area 2.a | 4.6 | 98 | - | 46 | **8.1** | 81 | **7.8** | 120 | 6.1 | 204 |
| Area 2.b | | **196** | | 83 | | 57 | | 43 | | 148 |
| Area 3 | 4.3 | **116** | - | 56 | **6.0** | - | 5.4 | 60 | **6.0** | 115 |
| Area 4.a | 3.9 | 69 | 2.3 | 56 | 5.3 | 49 | 4.3 | 52 | **6.6** | 136 |
| Area 4.b | | | | **133** | | 59 | | - | | 131 |
| Area 5 | 3.4 | 50 | 2.4 | 82 | 4.0 | 48 | 2.8 | 47 | 5.4 | **154** |
| Area 6 | 3.1 | - | 2.4 | - | **5.5** | 50 | 3.8 | - | **6.3** | 126 |
| Area 7 | 3.3 | - | 2.5 | - | **5.6** | 46 | 3.9 | 41 | **7.6** | 209 |

Notably, in agreement with the obtained results, both Type 2 and 3 events (Figure 12) are more severe than Type 1 in terms of coastal storms. Heavy rainfall produced local floods and dangerous discharges in ephemeral rivers during both Type 2 and Type 3 events. Nonetheless, Type 3 events are potentially the most compounding and intense, with Gloria 2020 being a perfect example of possible extreme impacts (e.g. Canals and Miranda, 2021; ICGC, 2021).

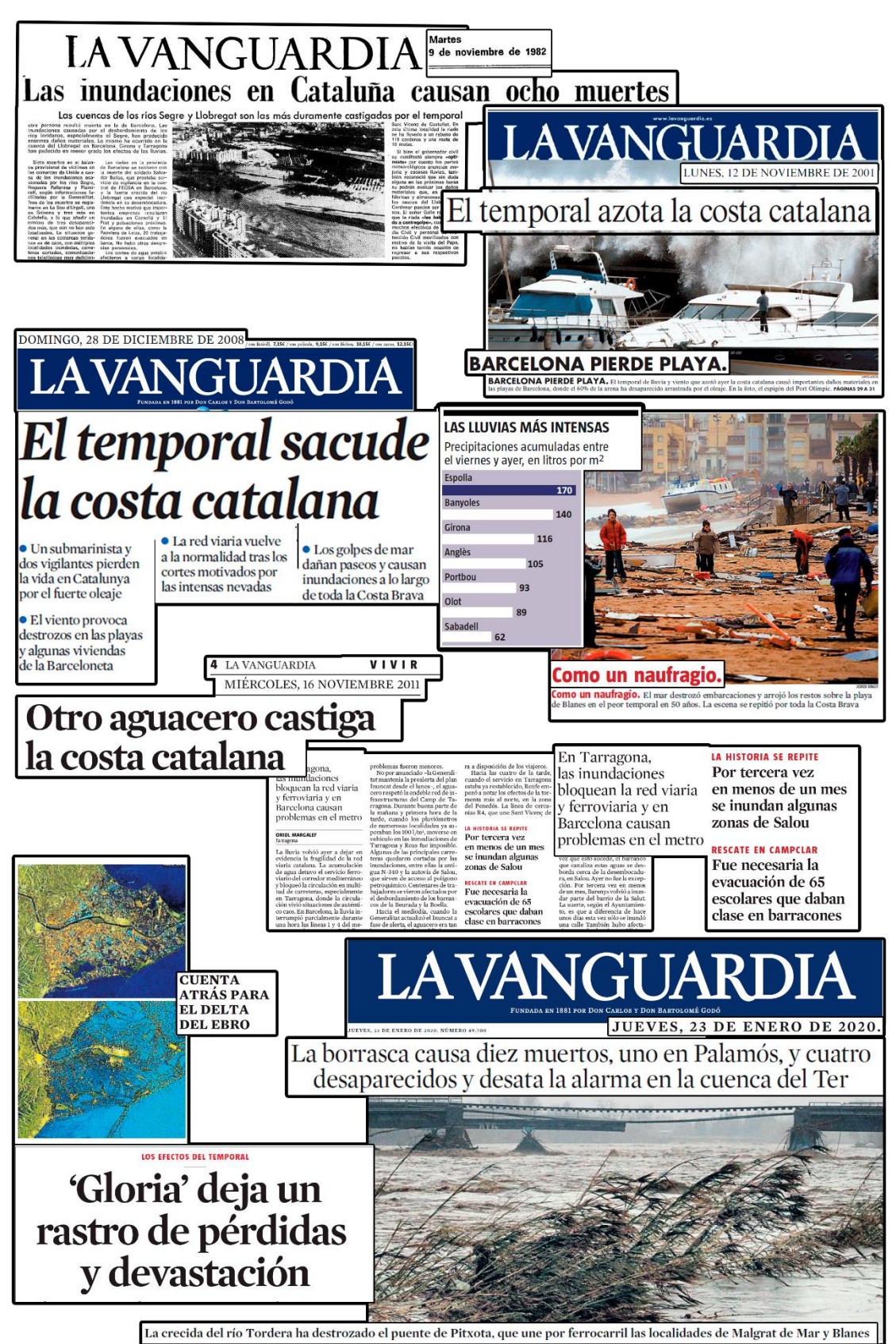

**Figure 13. Headlines in a local newspaper (La Vanguardia) after the impact of selected events (Figures 11 and 12). From top left to bottom right: (9/11/1982) "8 dead caused by floods in Catalonia" / "Segre and Llobregat basins, the most affected by the event"; (12/11/2001) "The storm hits the Catalan coast" / "Barcelona loses its beaches"; (28/12/2008) "The storm shakes the Catalan coast" / "The most intense rains" / "Like a wreck"; (16/11/2011) "The downpour punishes the Catalan coast" / " Flooding disrupts train and underground networks in Tarragona and Barcelona" / "Floods in Salou for the third time in a month" / "65 children evacuated from poorly-built schools"; (23/01/2020) "Gloria leaves a pathway of destruction and losses" / "The storm causes 10 dead (one in Palamós), 4 missing and all alarms triggered at the Ter basin" / "Final countdown at the Ebro Delta".**


## 5. Discussion

This study can be classified as an exploratory analysis prior to a classical probabilistic approach. While being simple, it permitted to identify the occurrence, main characteristics and spatial distribution of main types of compound events along the Catalan coastal zone at the NW Mediterranean. Due to specific conditions of the area, rainfall and waves are the drivers considered to compose the analysed events. The former is a proxy of runoff that results in flash floods and the latter is a proxy of run-up, which is dominant over storm-surge, while providing information on the magnitude of erosion processes.

The approach adopted to classify spatially compound events allows the identification of the dominant type of driver and, therefore, the dominant type of induced risks that will clearly condition risk management strategies. Moreover, in order to perform a sound bivariate probabilistic analysis of spatially compound events, it is necessary to define the spatial domain to be considered. In this sense, this preliminary exploratory analysis identifies the "connected" coastal sectors, and the dominant extreme contribution. Once identified, a more formal probabilistic analysis can be performed to calculate the probability of

occurrence of a given type of event in a given part of the territory.

This analysis has served to characterise the current scenario of these compound events in the NW Mediterranean on a time scale of about 40 years (1973 to 2013), which can be used as the reference state for future studies on the impacts of climate change. Obtained results show a spatial focus of most frequent co-occurrence and highest severity in the northernmost coast, as well as the absence of any statistically significant temporal trend in occurrence. With respect to future projections of

individual drivers, Tramblay and Somot (2018) report an increase in heavy rainfall in the northern Mediterranean basin, while Llasat et al (2016) report a possible increase in convective rainfall resulting in more flash-floods in the region. On the other hand, existing wave projections for the area do not show any statistically significant change in storminess (e.g. Casas-Prat and Sierra-Pedrico, 2013). In spite of this, future evolution of compound events will not necessarily be a linear combination of individual projections. At present, the existing information on the influence of climate change on compound events in the area

is limited to the analysis done by Bevacqua et al. (2019) at European scale, although they used storm surge as the marine component. The severity of induced damages, and the large spatial variation detected in their characteristics at a regional scale, make necessary to evaluate possible changes in their temporal and spatial occurrence, as well as in their intensity.

To implement the adopted methodological approach, a series of different choices were made that may condition the results obtained, which are discussed as follows. The basic spatial unit has been selected in terms of hydrological basins incorporating

all streams reaching the coastal zone in a given area, which in our case, were already defined by the Catalan Water Agency for hydrological management. The selection of automatic weather stations (AWS) was made to ensure good spatial and temporal coverages within basins along the coast during the study period (1973-2013). Although we have not performed a formal sensitivity analysis, the spatial coverage should ensure that significant heavy rainfall events will not be excluded even in spatially-localized episodes (in the case their scale is of the same order of magnitude of AWS local coverage). However, the

total number of AWS within a given basin could affect the maximum P24h value recorded for each event, as this value may spatially vary. Accordingly, although a change in the number of AWS could slightly affect the number of compound events when they are close to selected threshold conditions (P24h ~ 40 mm), and/or the rainfall peak value reached in a given basin, it is not expected to have a significant impact on the results obtained for assessment purposes.

In this work, we have used a three-days window to define compound events for consistency with the definition of coastal

storms in the study area. This is the time interval between consecutive storms to consider them statistically independent and generated by different meteorological conditions (e.g. Mendoza et al. 2011; Sanuy et al. 2020). When the time lag between consecutive storms is shorter than this value they are considered a multiple-peak event, which are not infrequent in the area and play an important role in controlling storm-induced coastal risk (see e.g. Sanuy and Jiménez 2021). Thus, heavy rainfall and wave storms occurring within this time window are part of the same event. Moreover, this time window is also meaningful

for risk management purposes, when in the presence of a SC-compound event, civil protection services may be overwhelmed when responding to cumulative impacts in spatially distant locations in the territory in such a short time interval. This value

depends on the characteristics of the study site, and the use of a different time window may be recommended in other areas depending on local (natural or management) conditions.

To characterize synoptic weather conditions responsible for analysed compound events, we have used data from NCEP/NCAR reanalysis. Although the relatively coarse resolution of this dataset will not permit to properly characterize mesoscale convective features, it is enough to represent the general synoptic conditions (e.g. Beck et al. 2016), and it has been used to investigate the relationship of different climate-related variables such as precipitation extremes, floods, river runoff and fires, with weather types in the NW Mediterranean basin (see e.g. Merino et al. 2016; Gilabert and Llasat, 2018; Duane and Brotons, 2019; Peña-Angulo et al. 2020). Although weather types have been classified using 1000 hPa geopotential, the convenience of incorporating upper-air through information has been reported when the final purpose is to predict rainfall (e.g. El Kenawy et al. 2014; Pook et al. 2014).

In this work we have identified synoptic types by using a correlation-based map classification, which is an intuitive and a simple way of automating the same task performed by an analyst (Yarnal, 1993 Yarnal et al., 2001). It produces good separation between weather types, i.e. a good degree of similarity between cases within the same cluster and dissimilarity between clusters (Huth et al., 2008). One of its main limitations is that is not as consistent as other approaches such as K-mean clustering or PCA, since it is generally sensitive to the choice of parameters to be set a priory (such as the cut-off threshold). This is also related to the fact the method tends to produce a large class followed by smaller ones (snowball effect). However, these limitations were minimized by performing a two-step inter-class comparison, i.e. a first classification with low thresholds (rt=0.2) and a second classification using the preliminary classes obtained in the first one and maximizing the correlation coefficient (rt), leading to final classes with two large groups (~40% of cases) and a follow-up one (~20% of cases). In any case, alternative weather typing could be implemented (e.g. Huth et al. 2008; Philipp et al. 2010; Dayan et al. 2012).

One of the important criteria applied to define the events was the spatial scale of the compounding effect. For the case of multivariate events, when both drivers must co-occur at the same site, the spatial dimension is here determined by the extent of the watersheds that collect rainfall discharging in a given area of the coastal zone. Other works, especially when dealing with large-scale analysis, such as Wahl et al (2015) and Ward et al (2018), establish the spatial link in terms of a maximum distance between rainfall and marine stations. While this is practical for identifying possible connected points at a very large scale, it is not necessarily physically correct. In the case of spatially compounding events, the scale is here defined in terms of risk management. In this sense, the maximum dimension of the area to compound the individual events is taken as the administrative region where the risks/damages should be managed by a given civil protection agency. This selection is based on the very reason underlying the definition of SC events, i.e. the potential overwhelming of the capacity of emergency-response services. Otherwise, it is very likely that if the spatial scale is extended, the probability of a spatially compound event will increase, although its individual induced impacts should not be managed together. In this context, the overall spatial scale of this study has been set to Catalonia, since the Catalan Government has the responsibility of managing Civil Protection services in this Autonomous region. Otherwise, from a climatological/physical standpoint, the area of analysis of potential spatially-connected events should be expanded to comprise the entire NW Mediterranean basin, where extreme precipitation events and coastal storms often impact in more than one "national" area (e.g. Lionello et al. 2006; Llasat et al. 2010; Raveh-Rubin and Wernli, 2015).

When analysing the importance of the different types of compound events along this part of the NW Mediterranean, on average, about 35 % of the events take place as multivariate, with the northernmost area being the area having the highest co-occurrence of up to 50 % of the events. This implies that, although they may be locally relevant, SC events are the most demanding in terms of risk management services. The most "extended" component across the territory during a compound event is the marine one, especially in the central part (areas 3 to 6) (figure 6) where, on average, the 67 % of the events present high waves. The exception to this is found in the northernmost areas 1 and 2, where the rainfall component slightly predominant, and in the southernmost zone, where both components are equally frequent. These areas at the limits of the territory are also where the

most intense components are found. Despite the spatial dominance of the marine component, the magnitude of damages across the territory is clearly dominated by extreme rainfall. The reason must be found in the scale of action of both components. Coastal storms impact on a fringe partially protected by beaches, with promenades and other linear infrastructures receiving most of the impact, in such a way that the extension of the hinterland to be affected is, in general, small and, in consequence, damages are limited to exposed values at these areas together the cost of recovery of beaches (e.g. Jiménez et al. 2011, 2012,

2018; Ballesteros et al. 2018a, 2018b; Sanuy and Jiménez, 2021). On the other hand, the occurrence of extreme rainfall in large areas within the catchment basin distributes the impact in a normally highly urbanized territory, as is the case of the Mediterranean coastal area, causing very large damages (e.g. Llasat et al. 2010; 2013; Barredo et al. 2012). This large difference between the magnitude of the impact of both components also conditions the main target of protection services that devote most of the efforts to manage rainfall/flood risks due to their greater severity.


## 6. Conclusions

From the obtained results, the north-western Mediterranean coast represented by the Catalan littoral zone can be characterised as an area with a relatively high probability of experiencing compound extreme events (3.4 events per year) as defined in terms of heavy rainfall (P24h) and wave storms (Hs). The most frequently found type along the territory is the spatially compound

event, which is mostly dominated by waves, whereas the influence of intense rainfall has a smaller spatial scale. However, even for the relatively small scale of the area (about 600 km of coastline), there is a significant variation in event characteristics along the territory, which ma y have important implications for risk management. Thus, the two northernmost sectors (Girona N and Lower Ter–Tordera) are the most likely to suffer from multivariate compound events, in such a way that they are the only geographical areas in which their frequency of occurrence exceeds the other type. These areas also present the highest

correlation in the intensity of both hazards (P24h and Hs). The other area in which multivariate events exceed the average frequency along the territory (although with a frequency smaller than spatially compound events) is the southernmost area (the lower Ebro and delta, Area 7).

This pattern is verified under all synoptic situations, although with some particularities that are related to dominant weather

conditions at the start of the compounding coastal storm. Thus, events generated under Type 1 conditions are dominated more by extreme rainfall because wave storms do not usually reach significantly high values, especially along the central part of the coast. In contrast, compound events generated under Types 2 and 3 are more likely to be characterised by the presence of extreme coastal storms, especially in the north, where they might also be accompanied by extreme rainfall (P24h > 100 mm). Type 2 events occur half as frequently as Types 1 and 3 and are mainly associated with the occurrence of extreme waves (Hs

> 5.5 m) at the northern and southern ends of the region. Under these synoptic situations, labelled as Mediterranean cyclones, the most extreme coastal storms have been recorded on the Catalan coast (Mendoza et al., 2011). Nonetheless, Type 3 events can be as severe as Type 2 events in terms of waves, with higher probabilities of compounding simultaneous extreme rainfall.

Compound event characteristics under each dominant weather type in terms of the spatial distribution and intensity were

characterised using a BN. With the exception of the two northernmost basins where multivariate events are dominant, the dominant typology is the spatially compound event (wave-dominated). This means that the extension of the affected area is usually larger for waves than for flash floods. In spite of this, the damage associated with heavy rainfall is usually much larger than that due to wave action.

The selected historical compound events are good examples of their potential consequences in an economically developed NW Mediterranean coastal zone. Even at a relatively small regional scale, they have an uneven spatial distribution in terms of the

dominant typology, hazard severity (rain and waves), and the correlation between them. The dominant synoptic conditions under which these events are generated have been clearly identified, with each inducing different types of events.


**Author contributions**

JAJ and MCL conceived the study. MS prepared the methodological framework and analysed the data, with all authors discussing results and implications. MS and TR were responsible for data pre-processing and curation. MS and JAJ prepared
the manuscript with contributions of all authors. JAJ and MCL were responsible for funding acquisition and supervision.

**Acknowledgements**

This work has been done in the framework of the M-CostAdapt (CTM2017-83655-C2-1&2-R) research project, funded by the Spanish Ministry of Economy and Competitiveness (MINECO/AEI/FEDER, UE). The authors express their gratitude to IH-
Cantabria and Puertos del Estado for supplying wave data, and AEMET and SMC for supplying rain data. Our thanks to Montserrat Llasat-Botija for her contribution in the identification of the compound events as well as for all the information about the impacts

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
