# Peer review of "Classifying compound coastal storm and heavy rainfall events in the north-western Spanish Mediterranean"

_Hydrology and Earth System Sciences, 2020_

## Referee Comment (RC1) · Anonymous Referee #1 · 8 Jan 2021

The paper describes an interesting study on the probability of occurrence of extreme combined events (floods and sea storms). The study clearly shows which are the types of synoptic patterns that generate the different types of event and identifies the areas where it is most likely in the occurrence of compound events (multivariate). I think the paper can be published with a few minor revisions. In particular I find some basic assumptions for the study not sufficiently justified, although I guess the valid justification is there. Specifically, 1) I would suggest explaining why it was assumed, to define an event as "compound" the time window of three days 2) Another assumption that should be more fully justified is why MSLP and geopotential at 1000 hpa were chosen to characterize the weather patterns. I also think that the final discussion can be extended,

introducing hints on how the two types of events identified (spatially compound and multivariate) influence the overall damage and also any risk management problems. For example, there is some evidence that floods and sea storms interact with each other during a multivariate event, aggravating the hazard scenario of one of the two (e.g. contemporary sea storm in the same area does increase the intensity of the flood hazard scenario ?)

---

## Referee Comment (RC2) · Jakob Zscheischler (Referee) · 11 Jan 2021

The authors present an interesting analysis of the relatively new concepts of multivariate and spatially compounding events considering the two hazards heavy precipitation and coastal storms along the Catalan coast. Overall the analysis is sound and fits well into HESS. However, some aspects of the methodology are difficult to follow (see comments below).

What is somewhat lacking is a discussion and contextualisation of the results. The current Discussion (L 425-568) can be considered as results and should be moved to the Results section (e.g. under the section headline "Case studies"). An actual discussion of the approach and results is missing. With the case studies, the manuscript is already quite extensive but maybe the authors could briefly discuss topics such as

- What are advantages/limitations of the approach the authors use to study compound events? How does it compare to other approaches in the literature?

- How would climate change affect the occurrence of this type of compound events in the study area?

Throughout the manuscript: check the usage of the word "verify". I think the word is used incorrectly and should always be replaced with "co-occur".

Minor comments:

Abstract:

- 3.4 events per year: difficult to contextualise if the definition of events is not presented. The number depends strongly on this definition.

- Last sentence: On what evidence is this conclusion based? Can you add this information here, please? Further, remove either "damage" or "impact" from the sentence (both mean the same here).

Main text:

L 76: "Spatially compounding events refer to co-occurring hazards from different climate drivers within a limited time window": maybe add "spatially" between before "co-occurring"

L 144: Is the analysis of spatially compounding events sensitive to the selection of AWS? I.e., if you include less/more stations, would this change your number of spatially compounding events?

L 187: Do you analyse the correlation between driver intensity in spatially compounding events? How? In those events you typically more than two variables. Please clarify.

L 201: Usage of "significant": consider using a different word (e.g. "extreme") since significant is usually only used in the context of statistical testing. Same comment applies to L 215.

L 227: add "heavy" before the second "rainfall"

L 235: I was at first confused about the usage of "areas". I assume you mean the areas delineated in Figure 2b. If this is the case, please make this clear (e.g. by referring to the figure). In L 237 you use the word "sector". Is this the same as the areas above? I assume it's a subset and you're referring only to the coastal areas. Please clarify.

L 237: The classification of compound events is unclear. Do you mean for each event you go through all the coastal sectors and check whether you have only rainfall or only wave extremes or both? Please make use of the word "extreme" to make clear what events you're talking about (e.g. instead of "episodes" in L 238). In particular, the phrase "where local extreme conditions correspond to..." is unclear. I think you mean something like "if rainfall/wave extremes occur in this sector. Also, the classes are not exclusive. An event can be multivariate, spatially compounding rain and spatially compounding waves. Which class wins?

L 242: See my comment higher up: what do you do when you have more then two drivers in the event?

L 294: remove "the presence of"

L 301 and 302: usage of "location": do you mean "area"?

L 305: it is not clear what the percentages in this paragraph refer to. Are they relative to all extreme event (i.e. 100% would mean all extreme events are compound events)? Please clarify.

L335: It seems that you pool all events in a given area even when they occur at different station. This should be mentioned in the methods section. It is still not clear how you deal with the case where multiple rainfall extremes in the same area co-occur with one

wave extreme.

L 358: "statistically independent values": please replace with "uncorrelated". A correlation of zero doesn't mean that the variables are statistically independent (though the reverse is true).

---

## Referee Comment (RC3) · Anonymous Referee #3 · 15 Jan 2021

This manuscript summarized results from a comprehensive analysis of compound coastal storms and heavy rainfall events in parts of Spain. Two types of compound events are assessed, namely multivariate compounding events and spatially compounding events. The analysis uses a mixture of observational data and model hindcasts, as well as atmospheric reanalysis data to assess synoptic weather types associated with certain compound events. The analysis and results are very interesting and worthy of publication with NHESS. A general comment is that the discussion could use more work as much of it reads more like results (this also relates to my comment below on what the impact of concern is when assessing for example spatially compounding events). Other than that I only have a list of mostly minor specific comments listed

below that should be taken into account before the paper is ready for publication.

37 Berghuijs et al.

40 The way it's worded indicates that Ward et al. considered storm surge and waves which is not true. Marcos et al. is a good reference for that and should be added: https://doi.org/10.1029/2019GL082599

56 I would refer to those as "regional" instead of local

75-82 What about spatially compounding with the same driver, i.e. spatial footprints (as analyzed here for example: https://doi.org/10.1029/2020JC016367), in my opinion that also falls into that category.

99 a bit more discussion about why events like that are of particular interest would be useful, what is the particular impact of concern for both types of events, multivariate or spatially compounding, in the context of this analysis?

165 Was there a particular reason to choose that reanalysis instead of a higher resolution one like ERA5? Do the authors expect all relevant features to be captured at this resolution?

184 "verify" is used several times in the wrong context

219-231 I understand that this would not include events where one variable is extreme and the other one is not (but might still be elevated, though not enough to cross the "extremes" threshold), is that correct? How does that relate to other approaches that are often used, such as two-sided sampling, where each extreme event of either variable is paired with the simultaneous value of the other variable (regardless whether the latter is extreme or not)

252 "correlation-based, gridded map-typing technique" is a mouthful and could use some further explanation.

265 More out of curiosity, do the authors have an idea how this compares to K-Means

clustering?

373 consider changing "generating floods" to "generating rainfall", as sea storms can also lead to floods

Caption of Table 3: should it be Figures 11 and 12?

---

## Author Response (AR1)

**Classifying compound coastal storm and heavy rainfall events in the north-western Spanish Mediterranean" by Marc Sanuy et al.**

**Anonymous Referee #1**

*We thank the reviewer for the detailed review and constructive comments on the manuscript. We have performed a thorough revision to address all the comments, as detailed below.*

The paper describes an interesting study on the probability of occurrence of extreme combined events (floods and sea storms). The study clearly shows which are the types of synoptic patterns that generate the different types of event and identifies the areas where it is most likely in the occurrence of compound events (multivariate). I think the paper can be published with a few minor revisions. In particular I find some basic assumptions for the study not sufficiently justified, although I guess the valid justification is there. Specifically,

1) I would suggest explaining why it was assumed, to define an event as "compound" the time window of three days.

*[R1.1] The use of a three-day window to define compound events was selected for consistency with the definition of coastal storms in the study area. This is the time span used between consecutive storms to consider them statistically independent and generated by different meteorological conditions (e.g. Mendoza et al. 2011; Sanuy et al. 2020). When the time lag between consecutive storms is shorter than this value they are considered to be a multiple-peak event, which are not infrequent in the area and play an important role in controlling storm-induced coastal risk (see e.g. Sanuy and Jiménez 2021). Thus, heavy rainfall and wave storms occurring within this time window are considered to be part of the same event. Moreover, this time window is short enough to be useful for analysing spatially-compound events (SC), when civil protection services may be overwhelmed by responding to cumulative impacts at spatially distant locations in the territory within such a short time interval. This explanation has been included in the discussion section, with a mention on the possible use of a different time window in other areas when local (natural or managerial) conditions recommend it.*

*Mendoza, E.T., Jiménez, J.A., and Mateo, J.: A coastal storms intensity scale for the Catalan sea (NW Mediterranean), Nat. Hazards Earth Syst. Sci., 11, 2453-2462, doi: 10.5194/nhess-11-2453-2011, 2011.*
*Sanuy M, Jiménez JA, Ortego MI, Toimil A., 2020. Differences in assigning probabilities to coastal inundation hazard estimators: Event versus response approaches. J Flood Risk Management, e12557. https://doi.org/10.1111/jfr3.12557*
*Sanuy, M., Jiménez, JA. 2021. Probabilistic characterisation of coastal storm-induced risks using Bayesian Networks. Natural Hazards & Earth System Sciences, 21, 219–238, doi: 10.5194/nhess-21-219-2021*

**NEW TEXT IN DISCUSSION:**

*In this work, we have used a three-days window to define compound events for consistency with the definition of coastal storms in the study area. This is the time interval between consecutive storms to consider them statistically independent and generated by different meteorological*

*conditions (e.g. Mendoza et al. 2011; Sanuy et al. 2020). When the time lag between consecutive storms is shorter than this value they are considered a multiple-peak event, which are not infrequent in the area and play an important role in controlling storm-induced coastal risk (see e.g. Sanuy and Jiménez 2021). Thus, heavy rainfall and wave storms occurring within this time window are part of the same event. Moreover, this time window is also meaningful for risk management purposes, when in the presence of a SC-compound event, civil protection services may be overwhelmed when responding to cumulative impacts in spatially distant locations in the territory in such a short time interval. This value depends on the characteristics of the study site, and the use of a different time window may be recommended in other areas depending on local (natural or management) conditions.*

2) Another assumption that should be more fully justified is why MSLP and geopotential at 1000 hpa were chosen to characterize the weather patterns.

[R1.2] The most usual analysis used for synoptic classifications is Sea Level Pressure (see for instance the special issue of the International Journal of Climatology on circulation types, Einar and Huth, 2016). In some cases, the 500hPa level is added in order to represent the configuration in the middle-high troposphere. Classic synoptic classifications such as Jenkinson and Collison (1977) were made from surface pressure maps since these could be constructed manually. However, today it is usual to analyze the synoptic configuration closest to the surface from the level of 1000 hPa since it represents well the behavior of the atmosphere at low levels. Surface pressure maps are used essentially for the location of fronts.

Jenkinson AF, Collison FP. 1977. An initial climatology of gales over the North Sea. Technical Report, Synoptic climatology Branch Memorandum No. 62, Meteorological Office, Bracknell, UK Einar, O., and R. Huth, 2016. Circulation-type classifications in Europe: results of the COST 733 Action. Int. J. Climatol. 36: 2671–2672 (2016). DOI: 10.1002/joc.4768 8 pp.

**MODIFICATIONS IN THE MANUSCRIPT**
In the previous version of the manuscript we were using MSLP maps only in Figures 11 and 12, to illustrate the specific events. Since it gives equivalent information to that of the 1000 hPa level and it has not been used for the weather typing, any use of MSLP has been removed from the manuscript to avoid confusion.

I also think that the final discussion can be extended, introducing hints on how the two types of events identified (spatially compound and multivariate) influence the overall damage and also any risk management problems. For example, there is some evidence that floods and sea storms interact with each other during a multivariate event, aggravating the hazard scenario of one of the two (e.g. contemporary sea storm in the same area does increase the intensity of the flood hazard scenario?)

[R1.3] This relates to [R2.1]. The Discussion section has undergone a thorough review. All text related to the description of specific episodes has been moved to a new subsection under Results. Additionally, the Discussion section now also addresses reviewers' comments and suggestions. With regards to the reviewer's specific comment, the two types of analysed compound events are compared in terms of their induced damages across the territory and how they can condition risk management operations. This has been done by comparing the observed consequences of the impact of selected events (as those reported in the paper -whose description will be moved to the Results section-). The reported Gloria January-2020 has been

used to illustrate the aggravation of inundation in low-lying coastal areas surrounding river mouths due to co-occurring high precipitation and storm waves. Thus, for instance, previous analysis of the impact of (univariate) extreme coastal storms in the Tordera river mouth area predicted significant flooding and erosion driven damages. However, these damages are limited to a relative narrow fringe along the coastline. The co-occurring high river discharges during the event in combination of increased water levels and waves at the river mouth resulted in a significant riverine inundation of the floodplain, together a large coastal reshaping. As a result of this, the extension of the inundation in the floodplain was significantly larger than the associated with the "univariate" storms. Alternatively, the consequences of spatially-compounding events have been illustrated with the co-occurring demand of civil protection services to manage storm-induced risks in different (and remote) parts of the territory, e.g. high precipitation and river discharges (evacuation in flood-prone areas) and extreme waves (damages in coastal infrastructures).

**NEW TEXT IN DISCUSSION:**

*One of the important criteria applied to define the events was the spatial scale of the compounding effect. For the case of multivariate events, when both drivers must co-occur at the same site, the spatial dimension is here determined by the extent of the watersheds that collect rainfall discharging in a given area of the coastal zone. Other works, especially when dealing with large-scale analysis, such as Wahl et al (2015) and Ward et al (2018), establish the spatial link in terms of a maximum distance between rainfall and marine stations. While this is practical for identifying possible connected points at a very large scale, it is not necessarily physically correct. In the case of spatially compounding events, the scale is here defined in terms of risk management. In this sense, the maximum dimension of the area to compound the individual events is taken as the administrative region where the risks/damages should be managed by a given civil protection agency. This selection is based on the very reason underlying the definition of SC events, i.e. the potential overwhelming of the capacity of emergency-response services. Otherwise, it is very likely that if the spatial scale is extended, the probability of a spatially compound event will increase, although its individual induced impacts should not be managed together. In this context, the overall spatial scale of this study has been set to Catalonia, since the Catalan Government has the responsibility of managing Civil Protection services in this Autonomous region. Otherwise, from a climatological/physical standpoint, the area of analysis of potential spatially-connected events should be expanded to comprise the entire NW Mediterranean basin, where extreme precipitation events and coastal storms often impact in more than one "national" area (e.g. Lionello et al. 2006; Llasat et al. 2010; Raveh-Rubin and Wernli, 2015).*

*When analysing the importance of the different types of compound events along this part of the NW Mediterranean, on average, about 35 % of the events take place as multivariate, with the northernmost area being the area having the highest co-occurrence of up to 50 % of the events. This implies that, although they may be locally relevant, SC events are the most demanding in terms of risk management services. The most "extended" component across the territory during a compound event is the marine one, especially in the central part (areas 3 to 6) (figure 6) where, on average, the 67 % of the events present high waves. The exception to this is found in the northernmost areas 1 and 2, where the rainfall component slightly predominant, and in the southernmost zone, where both components are equally frequent. These areas at the limits of the territory are also where the most intense components are found. Despite the spatial dominance of the marine component, the magnitude of damages across the territory is clearly dominated by extreme rainfall. The reason must be found in the scale of action of both components. Coastal storms impact on a fringe partially protected by beaches, with promenades and other linear infrastructures receiving most of the impact, in such a way that the extension of*

*the hinterland to be affected is, in general, small and, in consequence, damages are limited to exposed values at these areas together the cost of recovery of beaches (e.g. Jiménez et al. 2011, 2012, 2018; Ballesteros et al. 2018a, 2018b; Sanuy and Jiménez, 2021). On the other hand, the occurrence of extreme rainfall in large areas within the catchment basin distributes the impact in a normally highly urbanized territory, as is the case of the Mediterranean coastal area, causing very large damages (e.g. Llasat et al. 2010; 2013; Barredo et al. 2012). This large difference between the magnitude of the impact of both components also conditions the main target of protection services that devote most of the efforts to manage rainfall/flood risks due to their greater severity.*

**Jakob Zscheischler (Referee#2)**

jakob.zscheischler@climate.unibe.ch

We thank the reviewer for the detailed review and constructive comments on the manuscript. We have performed a thorough revision to address all the comments, as detailed below.

The authors present an interesting analysis of the relatively new concepts of multivariate and spatially compounding events considering the two hazards heavy precipitation and coastal storms along the Catalan coast. Overall the analysis is sound and fits well into HESS. However, some aspects of the methodology are difficult to follow (see comments below).

This is answered in the comments below.

What is somewhat lacking is a discussion and contextualisation of the results. The current Discussion (L 425-568) can be considered as results and should be moved to the Results section (e.g. under the section headline "Case studies"). An actual discussion of the approach and results is missing. With the case studies, the manuscript is already quite extensive but maybe the authors could briefly discuss topics such as:

[R2.1] As it was mentioned in [R1.3], the Discussion section has been fully modified. Following the reviewer's recommendation, all text describing specific compound episodes has been moved to the Results section. The final Discussion section now focuses on methods and results as the referee suggests, while also addressing comments of the other reviewers.

- What are advantages/limitations of the approach the authors use to study compound events? How does it compare to other approaches in the literature?

[R2.2] This study can be classified as an exploratory analysis prior to a classical probabilistic approach. While being rather simple, it permitted to identify the occurrence, main characteristics and spatial distribution of different types of compound coastal events at regional level, with rainfall and waves being the considered drivers; and to identify dominant weather types during such events. The adopted approach to classify spatially compound event permits to identify the dominant type of driver and, thus, the dominant type of induced risks which clearly will condition risk management strategies. Also, to perform a sound bivariate probabilistic analysis for spatially compound events it is necessary to define the spatial domain to be considered. In this sense, this previous exploratory analysis identifies "connected" coastal sectors, and the dominant extreme contribution. Once they are identified, a more formal probability analysis can be targeted to calculate the probability of occurrence of a given type of event in a given part of the territory.

This has been introduced in the discussion section along with a deeper contextualization of some of the specific methodological choices

**NEW TEXT IN DISCUSSION:**
*This study can be classified as an exploratory analysis prior to a classical probabilistic approach. While being simple, it permitted to identify the occurrence, main characteristics and spatial distribution of main types of compound events along the Catalan coastal zone at the NW Mediterranean. Due to specific conditions of the area, rainfall and waves are the drivers considered to compose the analysed events. The former is a proxy of runoff that results in flash*

*floods and the latter is a proxy of run-up, which is dominant over storm-surge, while providing information on the magnitude of erosion processes.*

*The approach adopted to classify spatially compound events allows the identification of the dominant type of driver and, therefore, the dominant type of induced risks that will clearly condition risk management strategies. Moreover, in order to perform a sound bivariate probabilistic analysis of spatially compound events, it is necessary to define the spatial domain to be considered. In this sense, this preliminary exploratory analysis identifies the "connected" coastal sectors, and the dominant extreme contribution. Once identified, a more formal probabilistic analysis can be performed to calculate the probability of occurrence of a given type of event in a given part of the territory.*

*[…]*

*In this work we have identified synoptic types by using a correlation-based map classification, which is an intuitive and a simple way of automating the same task performed by an analyst (Yarnal, 1993 Yarnal et al., 2001). It produces good separation between weather types, i.e. a good degree of similarity between cases within the same cluster and dissimilarity between clusters (Huth et al., 2008). One of its main limitations is that is not as consistent as other approaches such as K-mean clustering or PCA, since it is generally sensitive to the choice of parameters to be set a priory (such as the cut-off threshold). This is also related to the fact the method tends to produce a large class followed by smaller ones (snowball effect). However, these limitations were minimized by performing a two-step inter-class comparison, i.e. a first classification with low thresholds (rt=0.2) and a second classification using the preliminary classes obtained in the first one and maximizing the correlation coefficient (rt), leading to final classes with two large groups (~40% of cases) and a follow-up one (~20% of cases). In any case, alternative weather typing could be implemented (e.g. Huth et al. 2008; Philipp et al. 2010; Dayan et al. 2012).*

- How would climate change affect the occurrence of this type of compound events in the study area?

[R2.3] The purpose of this work is to characterize the current situation regarding the importance of analyzed compound events in the Spanish NW Mediterranean. This will be later (in a future work) used as a reference state to be compared with future scenarios to assess potential changes in probability of occurrence and/or intensity. At present, the existing information on the potential influence of climate change on compound coastal events in the area is limited to the analysis done by Bevacqua et al. 2019 at European scale. However, these authors analyzed rainfall and surge compound events (which are different drivers), they only considered multivariate co-occurring events, the scale of the work does not properly represent the regional dimension. This work is already cited in the manuscript and, now, we have included a specific comment in the Discussion section to stress the need to study future evolution of compound events in the area. In this sense, we have also referred to existing studies on future projection on individual drivers such as Tramblay and Somot (2018) who report an increase in intense rains in the north of the Mediterranean basin (Tramblay and Somot, 2018); and Llasat et al (2016) reporting a possible increase in convective rains that give rise to flash-floods in the region. Also, existing studies on the evolution of coastal storminess have been included. The underlying idea is to emphasize that, although studies on the projections of individual drivers show a given trend, future compound events scenarios will not necessarily be a linear combination of them.

Tramblay, Y., Somot, S. 2018. Future evolution of extreme precipitation in the Mediterranean. Climatic Change, 151:289–302 https://doi.org/10.1007/s10584-018-2300-5.

Llasat, M.C., R. Marcos, M. Turco, J. Gilabert, M. Llasat-Botija, 2016. Trends in flash flood events versus convective precipitation in the mediterranean region: the case of catalonia. Journal of Hydrology, 541, 24-37, http://dx.doi.org/10.1016/j.jhydrol.2016.05.040 0022-1694.

**NEW TEXT IN DISCUSSION:**

*This analysis has served to characterise the current scenario of these compound events in the NW Mediterranean on a time scale of about 40 years (1973 to 2013), which can be used as the reference state for future studies on the impacts of climate change. Obtained results show a spatial focus of most frequent co-occurrence and highest severity in the northernmost coast, as well as the absence of any statistically significant temporal trend in occurrence. With respect to future projections of individual drivers, Tramblay and Somot (2018) report an increase in heavy rainfall in the northern Mediterranean basin, while Llasat et al (2016) report a possible increase in convective rainfall resulting in more flash-floods in the region. On the other hand, existing wave projections for the area do not show any statistically significant change in storminess (e.g. Casas-Prat and Sierra-Pedrico, 2013). In spite of this, future evolution of compound events will not necessarily be a linear combination of individual projections. At present, the existing information on the influence of climate change on compound events in the area is limited to the analysis done by Bevacqua et al. (2019) at European scale, although they used storm surge as the marine component. The severity of induced damages, and the large spatial variation detected in their characteristics at a regional scale, make necessary to evaluate possible changes in their temporal and spatial occurrence, as well as in their intensity.*

Throughout the manuscript: check the usage of the word "verify". I think the word is used incorrectly and should always be replaced with "co-occur".

[R2.4] This has been addressed in the revised version of the manuscript.

Minor comments:

Abstract:

- 3.4 events per year: difficult to contextualise if the definition of events is not presented. The number depends strongly on this definition.

[R2.5] The text "(3.4 events per year)" ha been removed from the abstract.

- Last sentence: On what evidence is this conclusion based? Can you add this information here, please? Further, remove either "damage" or "impact" from the sentence (both mean the same here).

[R2.6] The last sentence has been rephrased to "Overall, results obtained from evidence from specific events indicated that heavy rainfall is related to the most significant impacts despite have damages having a larger spatial reach."

Main text:

L 76: "Spatially compounding events refer to co-occurring hazards from different climate drivers within a limited time window": maybe add "spatially" between before "cooccurring"

[R2.7] This has been addressed in the revised version of the manuscript. Now the sentence reads *"Spatially compounding events refer to co-occurring hazards from different climate drivers at distant locations within a limited time window"*:

L 144: Is the analysis of spatially compounding events sensitive to the selection of AWS? I.e., if you include less/more stations, would this change your number of spatially compounding events?

[R2.8] The selection of AWS was made to ensure good spatial and temporal coverages within basins during the studied period (1973-2013). Although we have not performed a formal sensitivity analysis, the spatial coverage should ensure that significant heavy rainfall events will not be excluded even in spatially-localized episodes. However, the total number of AWS within a given basin could affect the maximum P24h value recorded for each event, as this value may spatially vary. According to this, although a change in the number of AWS could slightly affect to number of compound events when they are close to threshold conditions, and/or the rainfall peak value reached within a given basin, it is not expected it will have a significant impact on obtained results for the purposes of the assessment. However, we have included a paragraph discussing the possible effects that spatial coverage of AWS within basing could have in our assessment, and in general, in any kind of assessment dealing with compounding events.

**NEW TEXT IN DISCUSSION:**
*To implement the adopted methodological approach, a series of different choices were made that may condition the results obtained, which are discussed as follows. The basic spatial unit has been selected in terms of hydrological basins incorporating all streams reaching the coastal zone in a given area, which in our case, were already defined by the Catalan Water Agency for hydrological management. The selection of automatic weather stations (AWS) was made to ensure good spatial and temporal coverages within basins along the coast during the study period (1973-2013). Although we have not performed a formal sensitivity analysis, the spatial coverage should ensure that significant heavy rainfall events will not be excluded even in spatially-localized episodes (in the case their scale is of the same order of magnitude of AWS local coverage). However, the total number of AWS within a given basin could affect the maximum P24h value recorded for each event, as this value may spatially vary. Accordingly, although a change in the number of AWS could slightly affect the number of compound events when they are close to selected threshold conditions (P24h ~ 40 mm), and/or the rainfall peak value reached in a given basin, it is not expected to have a significant impact on the results obtained for assessment purposes.*

L 187: Do you analyse the correlation between driver intensity in spatially compounding events? How? In those events you typically more than two variables. Please clarify.

[R2.9] The sentence has been rephrased to "*The results from (i) are used to assess the frequency of occurrence and spatial distribution of the different event types (multivariate and spatially compounding). At this stage, the correlation between driver intensity (i.e. the correlation between the maximum Hs and P24h) is also analysed for both event types.*"

In this study, we analyse both types of event based only on the two presented variables Hs and P24h

L 201: Usage of "significant": consider using a different word (e.g. "extreme") since significant is usually only used in the context of statistical testing. Same comment applies to L 215.

[R2.10] This has been addressed in the revised version of the manuscript

L 227: add "heavy" before the second "rainfall"

[R2.11] This has been addressed in the revised version of the manuscript

L 235: I was at first confused about the usage of "areas". I assume you mean the areas delineated in Figure 2b. If this is the case, please make this clear (e.g. by referring to the figure). In L 237 you use the word "sector". Is this the same as the areas above? I assume it's a subset and you're referring only to the coastal areas. Please clarify.

[R2.12] The words area and sector refer to the same thing, i.e. areas delineated in Figure 2.b. The text has been rephrased as follows:

*".... different characteristics along the costal basins (thereafter also named areas or sectors):"*

L 237: The classification of compound events is unclear. Do you mean for each event you go through all the coastal sectors and check whether you have only rainfall or only wave extremes or both? Please make use of the word "extreme" to make clear what events you're talking about (e.g. instead of "episodes" in L 238). In particular, the phrase "where local extreme conditions correspond to. . ." is unclear. I think you mean something like "if rainfall/wave extremes occur in this sector. Also, the classes are not exclusive. An event can be multivariate, spatially compounding rain and spatially compounding waves. Which class wins?

[R2.13] Indeed, the classes are not exclusive, and therefore, there is no winning class. The classification intends to classify the event as it is experienced at each basin. Thus, given a compound event (general) there will be basins experiencing it as multivariate (both components co-occur) and basins experiencing it as spatially compounding. In the second case, the basin can be receiving only rain (SC-rain) or only waves (SC-waves). By our definition of compound event, there will always be co-occurrence of the two analysed components at the regional scale (see next comment).

This part of the manuscript (right before section 3.4) has been rephrased to avoid confusion in our definition of compound event (regional scale), multivariate event (at the basin scale) or spatially compounding event (either waves or rain at the basin scale).

**NEW TEXT BEFORE SECTION 3.4**
*The classification intends to classify the event as it is experienced in each basin. Thus, in the face of a compound event (regional scale) there will be basins that experience it as multivariate (both components co-occur) and basins that experience it as spatially compounding. In the second case, the basin may be receiving only rain (SC-rain) or only waves (SC-waves). According to our definition of a compound event, there will always be a co-occurrence of the two components at the regional scale.*

L 242: See my comment higher up: what do you do when you have more then two drivers in the event?

[R2.14] Related to the former comment. By our definition of compound event, there will always be co-occurrence of the two analysed components at the regional scale (see previous comment). As an example, extreme rain at Area 1 co-occurring with extreme waves at Area 7 would be a spatially compound event. In Area 1 will be a spatially-compounding *rain*, and in Area 7 will be a spatially compounding *waves*. Another example could be extreme waves at all basins with rain in Areas 2 and 3. In this case Areas 2a and 3 would be experiencing a multivariate event, and all

other areas would be under spatially compounding waves. Notably, in all cases, both components co-occur at the regional scale.

The co-occurrence of extreme waves at different sectors (without rain present at any sector) is not studied here, as it would be a regular coastal storm, and not a compound (multi-hazard) event. The same is true for rain happening at multiple sectors without any extreme waves at the coast.

This part of the manuscript has been rephrased (see prior comment) to avoid confusion in our definition of compound event (regional scale), multivariate event (at the basin scale) or spatially compounding event (either waves or rain at the basin scale).

L 294: remove "the presence of"

[R2.15] This has been addressed in the revised version of the manuscript

L 301 and 302: usage of "location": do you mean "area"?

[R2.16] The word location has been changed to area or sector, as these where defined following [R2.12]

L 305: it is not clear what the percentages in this paragraph refer to. Are they relative to all extreme event (i.e. 100% would mean all extreme events are compound events)? Please clarify.

[R2.18] The percentages are relative to the defined compound events. This has been specified in the revised version of the manuscript

L335: It seems that you pool all events in a given area even when they occur at different station. This should be mentioned in the methods section. It is still not clear how you deal with the case where multiple rainfall extremes in the same area co-occur with one wave extreme.

[R2.18] If an event occurs at different stations of the same area, only the maximum P24h registered within the area is retained. The subjacent idea is to correlate the maximum P24h with the maximum wave, for each sector, and for each event. This was specified in the original manuscript L230 (*Finally, each compound event is characterised by the maximum P24h and Hs values of all stations and nodes within each coastal area during the event duration*). However, the sentence has been rephrased to avoid confusion or misunderstandings:: *Finally, each compound event is characterised by the maximum P24h and Hs values at the stations and nodes within each coastal area during the event duration*

L 358: "statistically independent values": please replace with "uncorrelated". A correlation of zero doesn't mean that the variables are statistically independent (though the reverse is true).

[R2.19] This has been addressed following reviewer's suggestion.

**Anonymous Referee #3**

We thank the reviewer for the detailed review and constructive comments on the manuscript. We have performed a thorough revision to address all the comments, as detailed below.

This manuscript summarized results from a comprehensive analysis of compound coastal storms and heavy rainfall events in parts of Spain. Two types of compound events are assessed, namely multivariate compounding events and spatially compounding events. The analysis uses a mixture of observational data and model hindcasts, as well as atmospheric reanalysis data to assess synoptic weather types associated with certain compound events. The analysis and results are very interesting and worthy of publication with NHESS. A general comment is that the discussion could use more work as much of it reads more like results (this also relates to my comment below on what the impact of concern is when assessing for example spatially compounding events). Other than that I only have a list of mostly minor specific comments listed below that should be taken into account before the paper is ready for publication.

[R3.1] As previously mentioned, the Discussion section has undergone a thorough review (see answers to R1.1 and R2.1).

37 Berghuijs et al.

[R3.2] This has been addressed in the revised version of the manuscript.

40 The way it's worded indicates that Ward et al. considered storm surge and waves which is not true. Marcos et al. is a good reference for that and should be added: https://doi.org/10.1029/2019GL082599

[R3.3] The sentence has been rephrased to "… by marine drives, such as waves and/or surge".

56 I would refer to those as "regional" instead of local

[R3.4] We now refer to them as "smaller regional scales".

75-82 What about spatially compounding with the same driver, i.e. spatial footprints (as analyzed here for example: https://doi.org/10.1029/2020JC016367), in my opinion that also falls into that category.

[R3.5] This aspect regards with the definition of compound event itself, and in spite that this situation could be relevant from the risk-management standpoint, here we consider this as a single coastal storm event (wave or surge) affecting a large part of the territory. In this study, we just consider compound events as those involving the co-occurrence of the two analysed components (rain and coastal -wave- storms) at the regional scale. (see answer to R2.14).

99 a bit more discussion about why events like that are of particular interest would be useful, what is the particular impact of concern for both types of events, multivariate or spatially compounding, in the context of this analysis?

[R3.6] The Discussion section has undergone a thorough review. All text related to the description of specific episodes will be moved to a new subsection under Results. Additionally,

the Discussion section now also addresses reviewers' comments and suggestions. With regards to the reviewer's specific comment, the two types of analysed compound events are compared in terms of their induced damages across the territory and how they can condition risk management operations. (see answer to comment R1.3)

165 Was there a particular reason to choose that reanalysis instead of a higher resolution one like ERA5? Do the authors expect all relevant features to be captured at this resolution?

[3.7] NCEP has been used as has been done in other articles (Wu et al., 2018). Since the main objective was to characterize synoptic weather conditions responsible for observed compound events, it is considered valid enough to represent the synoptic conditions that characterize them. It would not be the case, for instance, if you wanted to characterize mesoscale conditions. Examples of its application to characterize the types of circulation are in the special issue on circulation-type classifications (Einar and Huth, 2016), applications at the Mediterranean region by El Kenawy et al., (2014), Duane and Brotons (2019), Peña-Angulo et al., (2020) or Hochman ey al., (2020), and its application to the study area by Gilabert and Llasat (2018) or Lemus-Canovas et al (2019).

Additionally, it's easy to access, manage and download using programs in R.

El Kenawy, A. M., McCabe, M. F., Stenchikov, G. L., & Raj, J. (2014). Multi-decadal classification of synoptic weather types, observed trends and links to rainfall characteristics over Saudi Arabia. *Frontiers in Environmental Science*, *2*, 37. Doi: 10.3389/fenvs.2014.00037

Duane, A., & Brotons, L. (2018). Synoptic weather conditions and changing fire regimes in a Mediterranean environment. *Agricultural and Forest Meteorology*, *253*, 190-202.

Peña-Angulo, D., Nadal-Romero, E., González-Hidalgo, J. C., Albaladejo, J., Andreu, V., Bagarello, V., ... & Zorn, M. (2019). Spatial variability of the relationships of runoff and sediment yield with weather types throughout the Mediterranean basin. *Journal of Hydrology*, *571*, 390-405.

Hochman, A., Alpert, P., Kunin, P., Rostkier-Edelstein, D., Harpaz, T., Saaroni, H., & Messori, G. (2020). The dynamics of cyclones in the twentyfirst century: the Eastern Mediterranean as an example. *Climate Dynamics*, *54*(1), 561-574.

Lemus-Canovas, M., J.A. Lopez-Bustins, L. Trapero, J. Martin-Vide, 2019. Combining circulation weather types and daily precipitation modelling to derive climatic precipitation regions in the Pyrenees. Atmospheric Research 220 (2019) 181–193.

Einar, O., and R. Huth, 2016. Circulation-type classifications in Europe: results of the COST 733 Action. Int. J. Climatol. 36: 2671–2672 (2016). DOI: 10.1002/joc.4768 8 pp.

Gilabert, J. and M.C. Llasat, 2018. Circulation weather types associated with extreme flood events in Northwestern Mediterranean. Int. J. Climatol. (2018) Published online in Wiley Online Library (wileyonlinelibrary.com) DOI: 10.1002/joc.5301 Q1.

184 "verify" is used several times in the wrong context

[R3.8] This has been addressed in the revised version of the manuscript

219-231 I understand that this would not include events where one variable is extreme and the other one is not (but might still be elevated, though not enough to cross the "extremes" threshold), is that correct? How does that relate to other approaches that are often used, such

as two-sided sampling, where each extreme event of either variable is paired with the simultaneous value of the other variable (regardless whether the latter is extreme or not)

[R3.9] That's correct. Events where only one variable is extreme (it exceeds the threshold to be classified as extreme) are not included in the assessment. This approach has been adopted because this is mainly a risk-management oriented study. It is assumed that a given climatic variable (waves or rainfall) below the considered threshold is not producing a significant impact on the system by itself, neither its combination with an extreme one will substantially increase its associated risk. These events are considered as a single (univariate) extreme event. Their inclusion in the analysis would imply (in practice) to analyse all recorded univariate events and, thus, to substantially introduce noise in the analysis without providing significant information on compounding effects (which are the main target).

252 "correlation-based, gridded map-typing technique" is a mouthful and could use some further explanation.

[R3.10] This has been rephrased to: *"… a correlation-based method"* and later *"… the method consists on obtaining map patterns using the Pearson product-momentum correlation (rxy, eq.2) to depict the degree of similarity of spatial structures between pairs of gridded data (i.e. the map typing focuses on the positions of high- and low-pressure centres, rather than their magnitudes)"*.

265 More out of curiosity, do the authors have an idea how this compares to K-Means clustering?

[R3.11] The popularity of correlation-based map-pattern classification springs from its intuitive and simple basis of automating the same task performed manually by an analyst (Yarnal, 1993 Yarnal et al., 2001). Its product is easily read and understood by the user. It also produces a good separation between weather types, i.e., a good degree of similarity among the cases within the same cluster and dissimilarity between the clusters (Huth et al., 2008). One of its main limitations is that is not as consistent as other approaches such as K-mean clustering or PCA, as it is in general sensitive to the choice of parameters that must be set a priory (such as the cutting threshold). This is also related to the fact the method tends to produce one big class followed by minor ones (snowballing effect). However, these limitations where minimized by performing a two-step comparison between classes, i.e. a first classification with low thresholds (rt=0.2) and a second classification using the preliminary classes obtained in the first and maximizing the correlation coefficient (rt), which lead to the final classes with two big groups (~40% of cases) and a follow-up one (~20% of cases). On the contrary, one of the main limitations of a clustering-based method is its tendency to produce homogeneous (equally populated) groups.

The discussion section has been completed with a paragraph describing the advantages and limitations of the chosen method and how it compares with manual, cluster-based and PCA

**NEW TEXT IN DISCUSSION**
*In this work we have identified synoptic types by using a correlation-based map classification, which is an intuitive and a simple way of automating the same task performed by an analyst (Yarnal, 1993 Yarnal et al., 2001). It produces good separation between weather types, i.e. a good degree of similarity between cases within the same cluster and dissimilarity between clusters (Huth et al., 2008). One of its main limitations is that is not as consistent as other approaches such as K-mean clustering or PCA, since it is generally sensitive to the choice of parameters to be set a priory (such as the cut-off threshold). This is also related to the fact the*

*method tends to produce a large class followed by smaller ones (snowball effect). However, these limitations were minimized by performing a two-step inter-class comparison, i.e. a first classification with low thresholds (rt=0.2) and a second classification using the preliminary classes obtained in the first one and maximizing the correlation coefficient (rt), leading to final classes with two large groups (~40% of cases) and a follow-up one (~20% of cases). In any case, alternative weather typing could be implemented (e.g. Huth et al. 2008; Philipp et al. 2010; Dayan et al. 2012).*

373 consider changing "generating floods" to "generating rainfall", as sea storms can also lead to floods.

[R3.12] This has been addressed following reviewer's suggestion.

Caption of Table 3: should it be Figures 11 and 12?

[R3.13] Yes, it should. This has been addressed in the revised version of the manuscript.